# A comparative study of the cryo-EM structures of *Saccharomyces cerevisiae* and human anaphase-promoting complex/cyclosome (APC/C)

Ester Vazquez-Fernandez, Jing Yang, Ziguo Zhang, Antonina E Andreeva, Paul Emsley, David Barford*

MRC Laboratory of Molecular Biology, Cambridge, United Kingdom

*For correspondence: dbarford@mrc-lmb.cam.ac.uk

Competing interest: The authors declare that no competing interests exist.

**Abstract** The anaphase-promoting complex/cyclosome (APC/C) is a large multi-subunit E3 ubiquitin ligase that controls progression through the cell cycle by orchestrating the timely proteolysis of mitotic cyclins and other cell cycle regulatory proteins. Although structures of multiple human APC/C complexes have been extensively studied over the past decade, the *Saccharomyces cerevisiae* APC/C has been less extensively investigated. Here, we describe medium resolution structures of three *S. cerevisiae* APC/C complexes: unphosphorylated apo-APC/C and the ternary APC/C$^{CDH1}$-substrate complex, and phosphorylated apo-APC/C. Whereas the overall architectures of human and *S. cerevisiae* APC/C are conserved, as well as the mechanism of CDH1 inhibition by CDK-phosphorylation, specific variations exist, including striking differences in the mechanism of coactivator-mediated stimulation of E2 binding, and the activation of APC/C$^{CDC20}$ by phosphorylation. In contrast to human APC/C in which coactivator induces a conformational change of the catalytic module APC2:APC11 to allow E2 binding, in *S. cerevisiae* apo-APC/C the catalytic module is already positioned to bind E2. Furthermore, we find no evidence of a phospho-regulatable auto-inhibitory segment of APC1, that in the unphosphorylated human APC/C, sterically blocks the CDC20$^{C-box}$ binding site of APC8. Thus, although the functions of APC/C are conserved from *S. cerevisiae* to humans, molecular details relating to their regulatory mechanisms differ.

## eLife assessment

This study provides **compelling** data that defines the structure of the *S. cerevisiae* APC/C. The structure reveals overall conservation of its mechanism of action compared to the human APC/C but some **important** differences that indicate that activation by co-activator binding and phosphorylation are not identical to the human APC/C. Thus this study will be of considerable value to the field.

## Introduction

The anaphase-promoting complex/cyclosome (APC/C), through targeting specific proteins for proteolysis via the ubiquitin-proteosome system, is a key regulator of cell cycle transitions. APC/C activity and substrate selection are controlled at various levels to ensure that specific cell cycle events occur in the correct order and time, reviewed in *Alfieri et al., 2017*; *Barford, 2020*; *Bodrug et al., 2021*; *Watson et al., 2019a*. These regulatory mechanisms include the binding of cell-cycle-specific coactivator subunits (CDC20 and CDH1), reversible APC/C and coactivator phosphorylation, and inhibitory complexes and proteins.

Research on *Saccharomyces cerevisiae* APC/C over 25 years provided insights into APC/C structure and mechanism. Analysis of the sequences of proteins encoded by the CDC16, CDC23, and CDC27 genes led to the characterisation of the 34-residue tetratricopeptide repeat (TPR) motif that is usually arranged in multiple copies in contiguous arrays (*King et al., 1995*; *Lamb et al., 1994*; *Sikorski et al., 1991*). Similar arrangements of TPR motifs, that serve as protein-protein interaction sites, are conserved across a range of proteins with widely varying functions. The isolation of endogenous APC/C from *S. cerevisiae* confirmed the composition of large APC/C subunits that form a MDa-sized complex, and also the discovery of four small subunits (APC9, APC12/CDC26, APC13/SWM1, and APC15/MND2; *Hall et al., 2003*; *Passmore et al., 2003*; *Zachariae et al., 1998b*; *Zachariae et al., 1996*), that are also conserved with metazoan APC/C. *S. cerevisiae* APC/C was one of the first large multi-subunit complexes to be reconstituted in vitro using the baculovirus/insect cell over-expression system (*Schreiber et al., 2011*). Early electron microscopy studies of endogenous and recombinant *S. cerevisiae* APC/C revealed that a TPR module and a platform module assemble to create a triangular-shaped complex defining a central cavity which accommodates a combined substrate recognition module of the coactivator and APC10, and the catalytic module of APC2:APC11 (*da Fonseca et al., 2011*; *Schreiber et al., 2011*). Coactivator and APC10 cooperate to generate a co-receptor for D-box degron recognition (*Brown et al., 2015*; *Buschhorn et al., 2011*; *Carroll et al., 2005*; *Carroll and Morgan, 2002*; *Chang et al., 2014*; *Chao et al., 2012*; *da Fonseca et al., 2011*; *Hartooni et al., 2022*), whereas coactivator alone binds the KEN box and ABBA motif degrons (*Chao et al., 2012*; *He et al., 2013*; *Tian et al., 2012*).

Two coactivator subunits CDC20 and CDH1 determine APC/C substrate specificity throughout the mitotic cell cycle. The switching of these two coactivators regulates changes in APC/C substrate specificity. Additionally, coactivator-independent factors contribute to changes in substrate specificity (*Lu et al., 2014*). CDH1 interacts with the APC/C during G1, whereas CDH1 phosphorylation at the onset of S-phase inhibits its binding to the APC/C, a process that determines the irreversible transition from G1 into S-phase. On entering mitosis, CDK and polo kinases phosphorylate the APC/C to stimulate CDC20 binding to the APC/C (*Golan et al., 2002*; *Kraft et al., 2005*; *Kramer et al., 2000*; *Lahav-Baratz et al., 1995*; *Rudner and Murray, 2000*; *Shteinberg et al., 1999*; *Zachariae et al., 1998a*). In contrast to metazoan APC/C, *S. cerevisiae* APC/C utilises a third coactivator (Ama1) to regulate the events of meiosis (*Cooper et al., 2000*). As for human APC/C, *S. cerevisiae* APC/C functions using two E2s, a priming E2 that directly ubiquitylates APC/C substrates (Ubc4), and a processive E2 (Ubc1) that extends these ubiquitin moieties (*Rodrigo-Brenni et al., 2010*). Whereas the processive E2 that pairs with human APC/C (UBE2S) extends ubiquitin chains through K11 linkages (*Garnett et al., 2009*; *Jin et al., 2008*; *Matsumoto et al., 2010*; *Wickliffe et al., 2011*; *Williamson et al., 2009*), Ubc1 of *S. cerevisiae* synthesises K48-linked chains (*Rodrigo-Brenni et al., 2010*). Ubc4 and Ubc1 both bind the RING domain of APC11, and therefore compete for APC/C binding (*Girard et al., 2015*; *Rodrigo-Brenni and Morgan, 2007*). Ubc1 possesses an accessory UBA that enhances its affinity for the APC/C (*Girard et al., 2015*). The activities of both *S. cerevisiae* and human APC/C are inhibited by the mitotic checkpoint complex (MCC; *Burton and Solomon, 2007*; *Sudakin et al., 2001*), whereas Acm1 (*Martinez et al., 2006*) and EMI1 (early mitotic inhibitor 1; *Cappell et al., 2018*; *Reimann et al., 2001a*; *Reimann et al., 2001b*) are specific to *S. cerevisiae* and metazoan APC/C, respectively. These inhibitors exert their effects through a combination of pseudo-substrate motifs that block degron-binding sites on APC/C:coactivator complexes, and inhibition of E3 ligase catalytic activity by occluding E2 binding.

For our earlier research on the structure and mechanism of the APC/C, we investigated the *S. cerevisiae* system due to the ease of purifying TAP-tagged APC/C from yeast cultures (*Passmore et al., 2005*; *Passmore et al., 2003*), and at the time the benefits of yeast genetics had provided considerable insights into its function. By 2011 our capacity to over-express recombinant APC/C using the baculovirus insect cell system (*Schreiber et al., 2011*; *Zhang, 2016a*; *Zhang et al., 2013b*) allowed us and others to investigate human APC/C. These studies, together with those of others, provided insights into the overall architecture of the APC/C, mechanisms of substrate recognition and ubiquitylation, and APC/C regulation through mechanisms including reversible phosphorylation and the binding of the mitotic checkpoint complex and EMI1 (*Alfieri et al., 2016*; *Brown et al., 2015*; *Brown et al., 2016*; *Brown et al., 2014*; *Chang et al., 2014*; *Chang et al., 2015*; *Frye et al., 2013*; *Höfler et al., 2024*; *Watson et al., 2019b*; *Yamaguchi et al., 2016*; *Zhang et al., 2016b*).

Recently, we returned to *S. cerevisiae* APC/C to extend the 10 Å resolution structure published in 2011 (*da Fonseca et al., 2011*; *Schreiber et al., 2011*) to atomic resolution. We completed building a 4.0 Å structure of the *S. cerevisiae* APC/C^CDH1-substrate complex (APC/C^CDH1:Hsl1), allowing a comparative study of the *S. cerevisiae* and human systems. While human and *S. cerevisiae* APC/C architectures are in general very similar, there are significant differences, including the additional TPR subunit APC7 in human APC/C, and the structures of the smaller, less well-conserved subunits. Aspects of regulation also differ. For example, in contrast to human APC/C, the catalytic module APC2:APC11 of *S. cerevisiae* adopts an active conformation in the absence of coactivator, whereas in human APC/C coactivators induce a conformational change of APC2:APC11 from a downwards state, in which the E2-binding site is blocked, to an upwards state competent to bind E2 (*Chang et al., 2014*). Also unknown is the degree of conservation of the mechanism of APC/C^CDC20 activation by APC/C phosphorylation. In human APC/C, phosphorylation of an autoinhibitory (AI) segment incorporated within a long loop of the APC1 subunit removes a steric blockade to the binding of the CDC20 C-box to its binding site on APC8 (*Fujimitsu et al., 2016*; *Qiao et al., 2016*; *Zhang et al., 2016b*). In our *S. cerevisiae* apo-APC/C cryo-EM maps we observe no evidence of an auto-inhibitory segment bound to the coactivator C box-binding site in APC8. Thus it seems likely that mechanisms of activation of APC/C^CDC20 by phosphorylation are not conserved from *S. cerevisiae* to human. Here, we present a comparative study of human and *S. cerevisiae* APC/C.

## Results
### Overall structure of *S. cerevisiae* APC/C complexes

We reconstituted *S. cerevisiae* APC/C using the baculovirus/insect cell system (*Zhang et al., 2016c*; *Figure 1—figure supplement 1A*), similar to methods described previously (*Schreiber et al., 2011*). When in complex with the CDH1 coactivator, the reconstituted APC/C ubiquitylated the high-affinity *S. cerevisiae* substrate Hsl1 (*Figure 1—figure supplement 1B, C*). Phosphorylated APC/C was generated using CDK2-cyclin A, and mass spectrometry revealed the majority of phosphosites mapped to the APC1, APC3 and APC6 subunits (*Supplementary file 1a*). We prepared cryo-EM grids for three APC/C complexes; (i) a ternary APC/C^CDH1:Hsl1 complex, (ii) apo-APC/C and (iii) phosphorylated apo-APC/C. A cryo-EM reconstruction of the ternary APC/C^CDH1:Hsl1 complex was determined at 4 Å resolution (*Figure 1*; *Figure 1—figure supplements 2 and 3* and *Table 1*), whereas the structures of unphosphorylated and phosphorylated apo-APC/C were determined at 4.9 Å and 4.5 Å resolution, respectively (*Figure 2* and *Figure 2—figure supplements 1 and 2*, *Figure 3* and *Figure 3—figure supplements 1 and 2*, and *Table 1*). We built the atomic coordinates based on prior structures of human (*Chang et al., 2015*; *Höfler et al., 2024*), *E. cuniculi* (*Zhang et al., 2010b*), *S. cerevisiae* (*Au et al., 2002*), and *S. pombe* (*Zhang et al., 2013a*; *Zhang et al., 2010a*) APC/C subunits, and used AlphaFold2 predictions (*Jumper et al., 2021*; *Tunyasuvunakool et al., 2021*) to guide de novo model building (*Supplementary file 1b*). The overall structure of the recombinant APC/C^CDH1:Hsl1 ternary complex is similar to the 10 Å resolution cryo-EM structure of APC/C^CDH1:Hsl1 determined using native APC/C purified from endogenous sources (*da Fonseca et al., 2011*), and our earlier negative stain EM reconstruction (*Schreiber et al., 2011*). The complex adopts a triangular structure delineated by a lattice-like shell that generates a central cavity (*Figure 1*). The APC/C is comprised of two major modules; the TPR lobe composed of the canonical TPR subunits APC3/CDC27, APC6/CDC16 and APC8/CDC23, and the platform module composed of the large APC1 subunit, together with APC4 and APC5. The TPR lobe adopts a quasi-symmetric structure through the sequential stacking of the three structurally related TPR homo-dimers, APC8/CDC23, APC6/CDC16 and APC3/CDC27. Each TPR protein is composed of 12–13 copies of the TPR motif, a 34-residue repeat that forms a pair of anti-parallel α-helices. Consecutive arrays of TPR motifs fold into a TPR superhelical structure with a pitch of seven TPR motifs (*Das et al., 1998*). The TPR helix defines an inner TPR groove suitable for protein binding. Each of the three TPR proteins of the TPR lobe homo-dimerise through their N-terminal TPR helix. The two C-terminal TPR helices of the APC3 homo-dimer provide the IR-tail binding site for coactivators and APC10 (*Chang et al., 2014*; *Chang et al., 2015*; *Matyskiela and Morgan, 2009*), whereas structurally equivalent sites on APC8 interacts with the C-box of coactivator N-terminal domains (NTDs) (*Chang et al., 2015*), and also with the IR tail of the the CDC20 subunit of the mitotic checkpoint complex (MCC^CDC20) of the APC/C^MCC complex (*Alfieri et al., 2016*; *Yamaguchi*

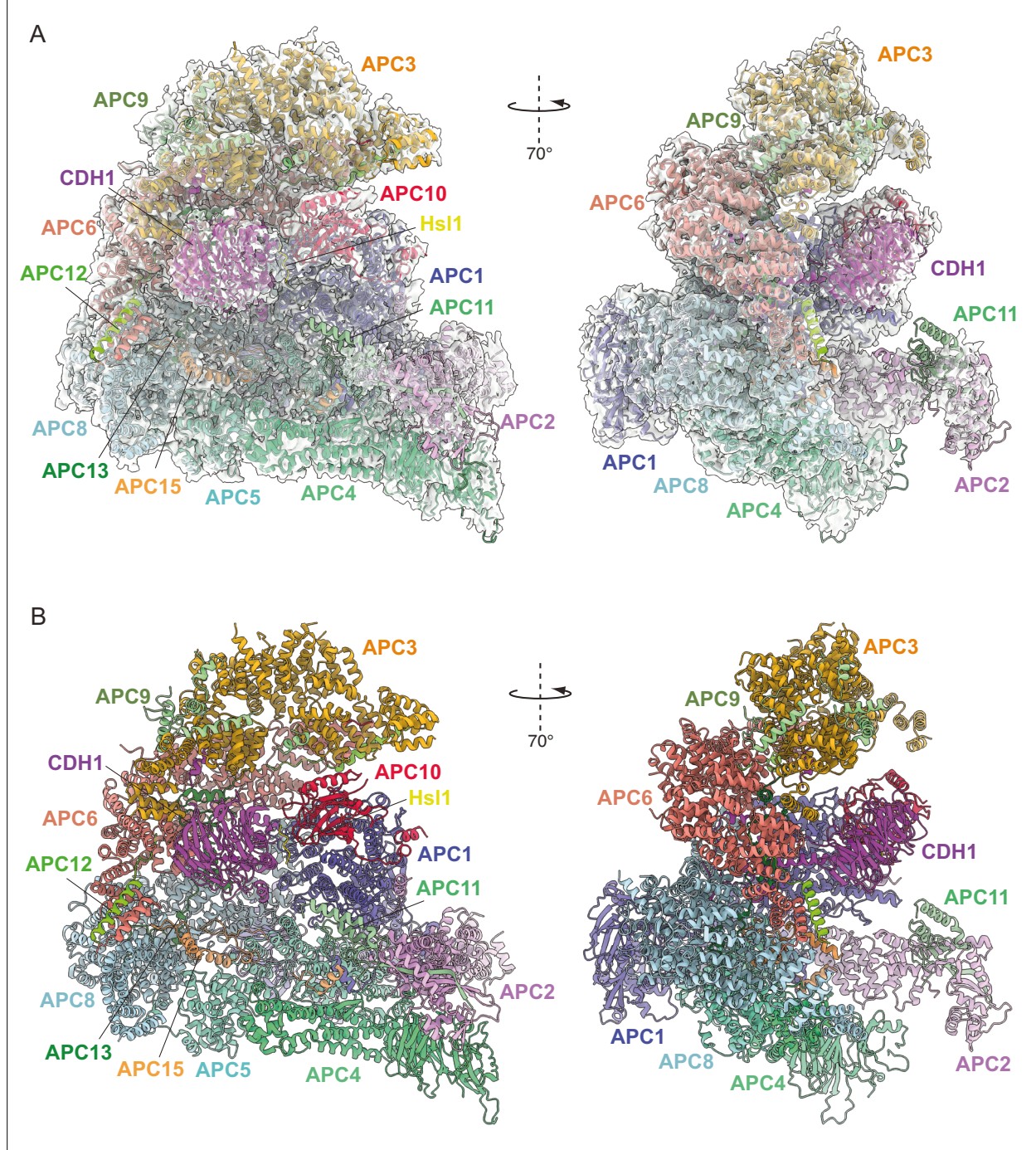

**Figure 1.** Overall structure of the APC/C$^{CDH1:Hsl1}$ complex. (**A**) Two views of the APC/C$^{CDH1:Hsl1}$ ternary complex fitted into the 4.0 Å cryo-EM map. (**B**) Two views of the APC/C$^{CDH1:Hsl1}$ ternary complex shown as ribbon representations.

The online version of this article includes the following source data and figure supplement(s) for figure 1:

**Figure supplement 1.** SDS PAGE gels of purified APC/C complexes and ubiquitylation assays.

**Figure supplement 1—source data 1.** PDF file of original Coomassie stained SDS PAGE gel for *Figure 1—figure supplement 1A* and Western blots for *Figure 1—figure supplement 1B and C*, with no labels.

**Figure supplement 1—source data 2.** PDF file of original Coomassie stained SDS PAGE gel for *Figure 1—figure supplement 1A* and Western blots for *Figure 1—figure supplement 1B and C*, indicating relevant lanes (boxed in red) and treatments.

**Figure supplement 2.** EM images and 2D class averages of APC/C$^{CDH1:Hsl1}$ complex.

**Figure supplement 3.** Data processing pipeline for APC/C$^{CDH1:Hsl1}$ complex cryo-EM reconstructions.

**Table 1.** Cryo-EM data collection, refinement, and validation statistics.

| | Apo-APC/C (EMD-15199) (PDB 8A5Y) | Phosphorylated apo-APC/C (EMD-15201) (PDB 8A61) | APC/C$^{CDH1:Hsl1}$ (EMD-15123) (PDB 8A3T) |
|---|---|---|---|
| **Data collection** | | | |
| EM | FEI Titan Krios | FEI Titan Krios | FEI Titan Krios |
| Detector | FEI Falcon III | FEI Falcon III | FEI Falcon III |
| Magnification | 59,000 | 59,000 | 59,000 |
| Voltage (keV) | 300 | 300 | 300 |
| Electron exposure (e-/Å$^2$) | 59 | 59 | 59 |
| Defocus range (μm) | 2.6–3.9 | 2.6–3.9 | 2.6–3.9 |
| Pixel size (Å) | 1.38 | 1.38 | 1.38 |
| **Reconstruction** | | | |
| Software | RELION 3.1 | RELION 3.1 | RELION 3.1 |
| Symmetry imposed | C1 | C1 | C1 |
| Initial particle images (N) | 815,009 | 806,068 | 1,425,386 |
| Final particle images (N) | 268,102 | 200,310 | 249,193 |
| Accuracy of rotations (°) | 1.76 | 1.79 | 1.16 |
| Accuracy of translations (°) | 1.01 | 0.93 | 0.62 |
| Map resolution (Å) | 4.9 | 4.4 | 4.0 |
| FSC threshold | 0.143 | 0.143 | 0.143 |
| **Refinement** | | | |
| Software | Phenix | | |
| Resolution limit (Å) | 4.9 | 4.4 | 4.0 |
| RMSD bond length (Å) | 0.004 | 0.003 | 0.004 |
| RMSD bond angle (°) | 0.878 | 0.641 | 0.754 |
| *Model to map fit* | | | |
| CC_mask | 0.756 | 0.793 | 0.752 |
| CC_volume | 0.748 | 0.785 | 0.750 |
| **Validation** | | | |
| All-atom clash score | 18.46 | 15.14 | 18.84 |
| Ramachandran plot | | | |
| Preferred (%) | 95.76 | 95.74 | 95.93 |
| Allowed (%) | 3.94 | 4.02 | 3.90 |
| Outliers (%) | 0.34 | 0.24 | 0.17 |

*et al., 2016*). Assembly of the complete APC/C complex is augmented by four small non-globular intrinsically disordered proteins (IDPs) that by interacting simultaneously with multiple large globular subunits stabilise the APC/C.

Together, the platform and TPR modules create a scaffold for the juxtaposition of the catalytic and substrate recognition modules, with the catalytic module of the cullin subunit APC2 and the RING-domain subunit APC11 binding to the platform module. Two subunits mediate substrate recognition through a substrate-recognition module: the exchangeable coactivator subunits (CDC20, CDH1, and Ama1) and the core APC/C subunit APC10. Coactivators and APC10 share in common a conserved

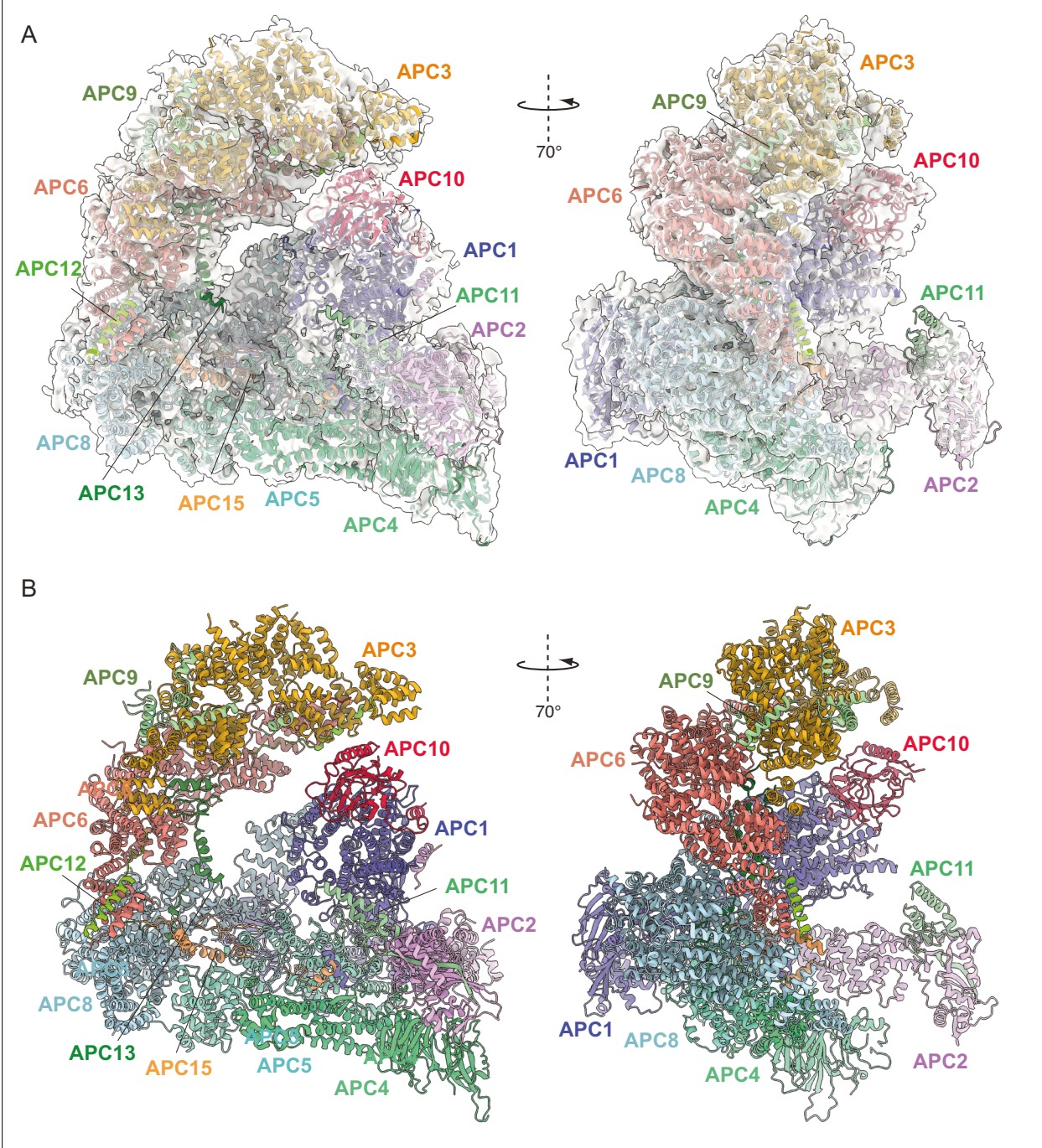

**Figure 2.** Overall structure of unphosphorylated apo-APC/C. (**A**) Two views of apo-APC/C fitted into the 4.9 Å cryo-EM map. (**B**) Two views of apo-APC/C shown as ribbon representations.

The online version of this article includes the following figure supplement(s) for figure 2:

**Figure supplement 1.** Cryo-EM images and 2D class averages of apo-APC/C complexes.

**Figure supplement 2.** Data processing pipeline for unphosphorylated apo-APC/C cryo-EM reconstructions.

C-terminal IR (Ile-Arg) tail that each interacts with the symmetry-related subunits of the APC3 homodimer. Additionally, APC10 interacts with APC1, and the NTDs of coactivators interact with both the TPR and platform modules. Cryo-EM density bridging the CDH1 WD40 domain (CDH1$^{WD40}$) and APC10 corresponds to the D-box degron of Hsl1 (*da Fonseca et al., 2011*). In all three states, the APC2:APC11 catalytic module adopts an upward conformation that positions the APC11 RING

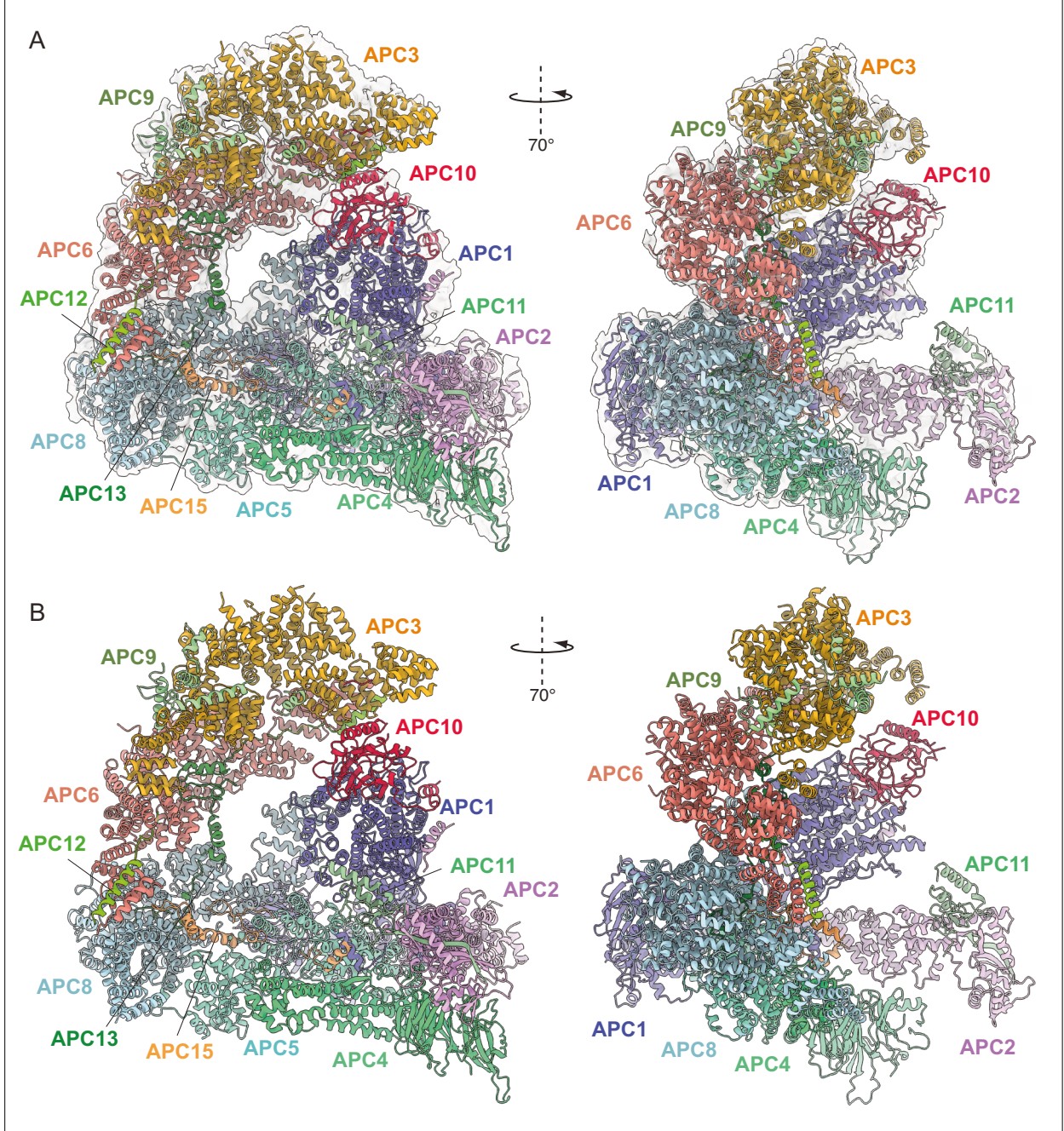

**Figure 3.** Overall structure of phosphorylated apo-APC/C. (**A**) Two views of phosphorylated apo-APC/C fitted into the 4.5 Å cryo-EM map. (**B**) Two views of apo-APC/C shown as ribbon representations.

The online version of this article includes the following figure supplement(s) for figure 3:

**Figure supplement 1.** Cryo-EM images and 2D class averages of phosphorylated apo-APC/C complexes.

**Figure supplement 2.** Data processing pipeline for phosphorylated apo-APC/C cryo-EM reconstructions.

domain (APC11$^{RING}$) proximal (~30 Å) to the substrate (*Figures 1–3*), similar to that observed in the active human APC/C$^{CDH1:EMI1}$ complex (*Figure 4A and B*; *Chang et al., 2015*; *Höfler et al., 2024*).

## Detailed subunit analysis

### APC2

A recent analysis of human APC/C$^{CDH1:EMI1}$ determined at 2.9 Å resolution revealed a zinc-binding module (APC2$^{ZBM}$) inserted within cullin repeat 2 (CR2) of APC2 (*Höfler et al., 2024*). This segment constitutes a 45-residue insert between the A and B α-helices of CR2 (*Figure 5A*), and is most similar in structure to treble-clef/GATA-like zinc fingers. Four Cys residues coordinate the zinc ion in human APC2. A multiple sequence alignment showed that APC2$^{ZBM}$ is conserved within metazoan APC2 sequences (from humans to *C. elegans*), but not in yeast APC2 sequences (*Chang et al., 2015*; *Höfler et al., 2024*). Consistent with this analysis, there is no structural equivalent of human APC2$^{ZBM}$ in the cryo-EM structure of *S. cerevisiae* APC/C$^{CDH1:Hsl1}$ (*Figure 5B and C*). APC2$^{ZBM}$ substantially increases the thermal stability of human APC2. Whether it performs additional roles is unknown.

Similarly to human APC/C structures, the C-terminal domain (CTD) of APC2 (APC2$^{CTD}$), together with the associated APC11 subunit, are less well defined than other regions of the complex, with the WHB domain of APC2 (APC2$^{WHB}$) being highly flexible and not visible in cryo-EM density (*Figure 1A* and *Figure 1—figure supplement 2E*). In human APC/C structures, APC2$^{WHB}$ is likewise flexible when not in complex with the priming E2 (UbcH10) or bound to the MCC (*Brown et al., 2015*; *Chang et al., 2015*). APC2$^{WHB}$ becomes ordered when the APC/C forms complexes with either UbcH10 (*Brown et al., 2015*), the MCC (*Alfieri et al., 2016*; *Yamaguchi et al., 2016*), Nek2A (*Alfieri et al., 2020*), or SUMOylated (*Yatskevich et al., 2021*).

### APC4

The APC4 subunit comprises a WD40 domain with a long four-helix bundle insert between blades 3 and 4, augmented by an insert within blade 3 that is an α-helix in human APC4 and a β-hairpin in *S. cerevisiae* APC4 (*Figure 5D and E*). Strikingly, in contrast to human APC4 which incorporates a typical seven-β-bladed propeller, in *S. cerevisiae* the APC4 WD40 domain (APC4$^{WD40}$) is constructed from six blades (blades 1–6; *Figure 5D*). The absence of cryo-EM density for a blade7 in *S. cerevisiae* APC4 is supported by an AlphaFold2 prediction (*Tunyasuvunakool et al., 2021*; *Figure 5F and G*). The predicted APC4 model includes an α-helix instead of blade7. However, this helix is not supported by corresponding cryo-EM density, and is a low confidence prediction based on a low pLDDT score (*Figure 5F and G*). The unusual APC4$^{WD40}$ domain architecture might be specific to *S. cerevisiae*. The AlphaFold2 prediction of *S. pombe* APC4 proposes a seven-bladed WD40 domain (*Tunyasuvunakool et al., 2021*).

### Other platform module subunits

APC1: APC1, the largest APC/C subunit (1748 residues in *S. cerevisiae*), is composed of an N-terminal WD40 β-propeller domain (APC1$^{WD40}$) and a toroidal PC domain (proteosome/cyclosome; APC1$^{PC}$) interconnected by a predominantly α-helical solenoid domain (APC1$^{Mid}$). APC5 comprises an N-terminal α-helical domain and C-terminal TPR domain of 13 TPR motifs generating two TPR helical turns. APC10 is a DOC homology domain comprising a β-sandwich homologous to galactose-binding domains (*Au et al., 2002*; *Wendt et al., 2001*).

## Structure of intrinsically disordered regions and intrinsically disordered proteins

Our structure indicates how the four smaller non-globular subunits (APC9, CDC26/APC12, SWM1/APC13, MND2) (*Hall et al., 2003*; *Passmore et al., 2003*; *Zachariae et al., 1998b*; *Zachariae et al., 1996*), with a high content of intrinsically disordered regions (IDRs), mediate inter-subunit interactions between the larger globular subunits (*Figure 4*). CDC26/APC12 and SWM1/APC13 share sequence conservation with their metazoan counterparts (*Schwickart et al., 2004*; *Supplementary file 1b*), consistent with the proposal that SWM1 is the ortholog of metazoan APC13 (*Hall et al., 2003*; *Passmore et al., 2003*). MND2 and APC9 share no clear sequence similarities with metazoan APC/C subunits, although previous structure-based mapping of MND2 and APC9 to the *S. cerevisiae*

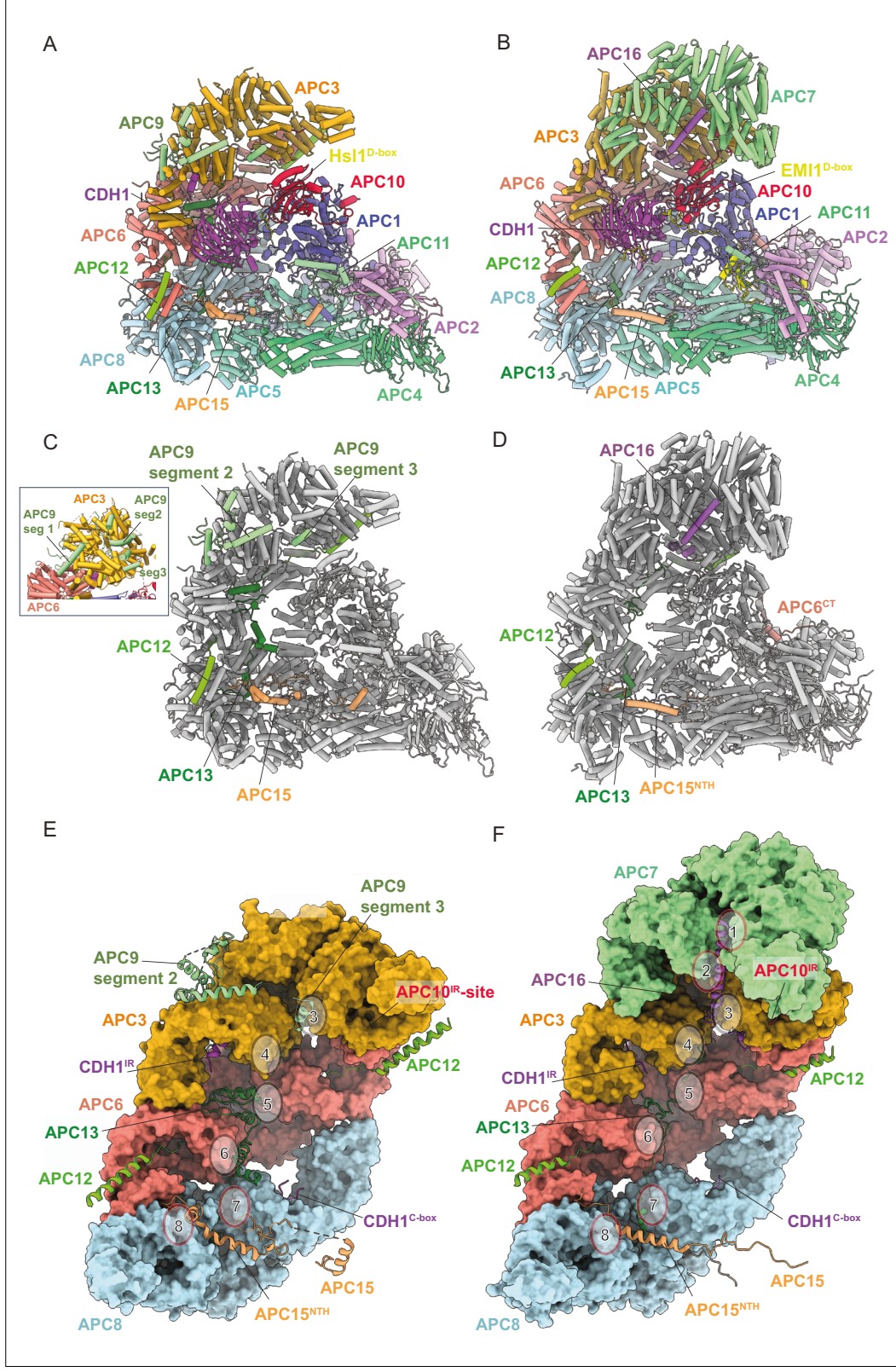

**Figure 4.** Comparison of *S. cerevisiae* APC/C$^{CDH1:Hsl1}$ with human APC/C$^{CDH1:EMI1}$. (**A**) *S. cerevisiae* and (**B**) human complexes subunits colour-coded. (**C**) *S. cerevisiae* and (**D**) human complexes with large subunits coloured in grey and small IDP subunits (APC9, APC12, APC13, APC15, APC16) colour-coded. The structure of human APC/C$^{CDH1:EMI1}$ from *Höfler et al., 2024* (PDB 7QE7). (**E**) and (**F**) four small IDP-subunits (APC9, APC13, APC15, APC16) contribute

*Figure 4 continued on next page*

*Figure 4 continued*

to interacting with equivalent, quasi-symmetrical sites on the outer surfaces of the TPR subunits of the TPR lobe. TPR lobes of *S. cerevisiae* (**E**) and human APC/C (**F**) are depicted as a surface representations. The contact sites with the three small subunits that contact the *S. cerevisiae* TPR lobe are numbered 3–8 after human APC/C (*Chang et al., 2015*) that has sites 8 sites due to APC7.

APC/C EM-reconstruction suggested counterparts to APC15 and APC16, respectively (*Schreiber et al., 2011*), consistent with the proposal that MND2 is the *S. cerevisiae* ortholog of human APC15 (*Mansfeld et al., 2011*), and that MND2 and APC15 share related functions in promoting the auto-ubiquitylation of CDC20 in the context of the *S. cerevisiae* and human APC/C:MCC complexes (*Alfieri et al., 2016*; *Foster and Morgan, 2012*; *Mansfeld et al., 2011*; *Uzunova et al., 2012*). Analysis of the subunit composition of APC/C complexes purified from *S. cerevisiae* strains harbouring specific gene deletions revealed that APC9 is required for the efficient incorporation of CDC27/APC3 into the assembled APC/C (*Passmore et al., 2003*; *Zachariae et al., 1998b*), whereas SWM1/APC13 and CDC26/APC12 are required for the stoichiometric assembly of APC3, APC6 and APC9 into the complex (*Schwickart et al., 2004*; *Zachariae et al., 1998b*). MND2 deletion did not affect the association of other subunits (data not shown from *Schwickart et al., 2004*).

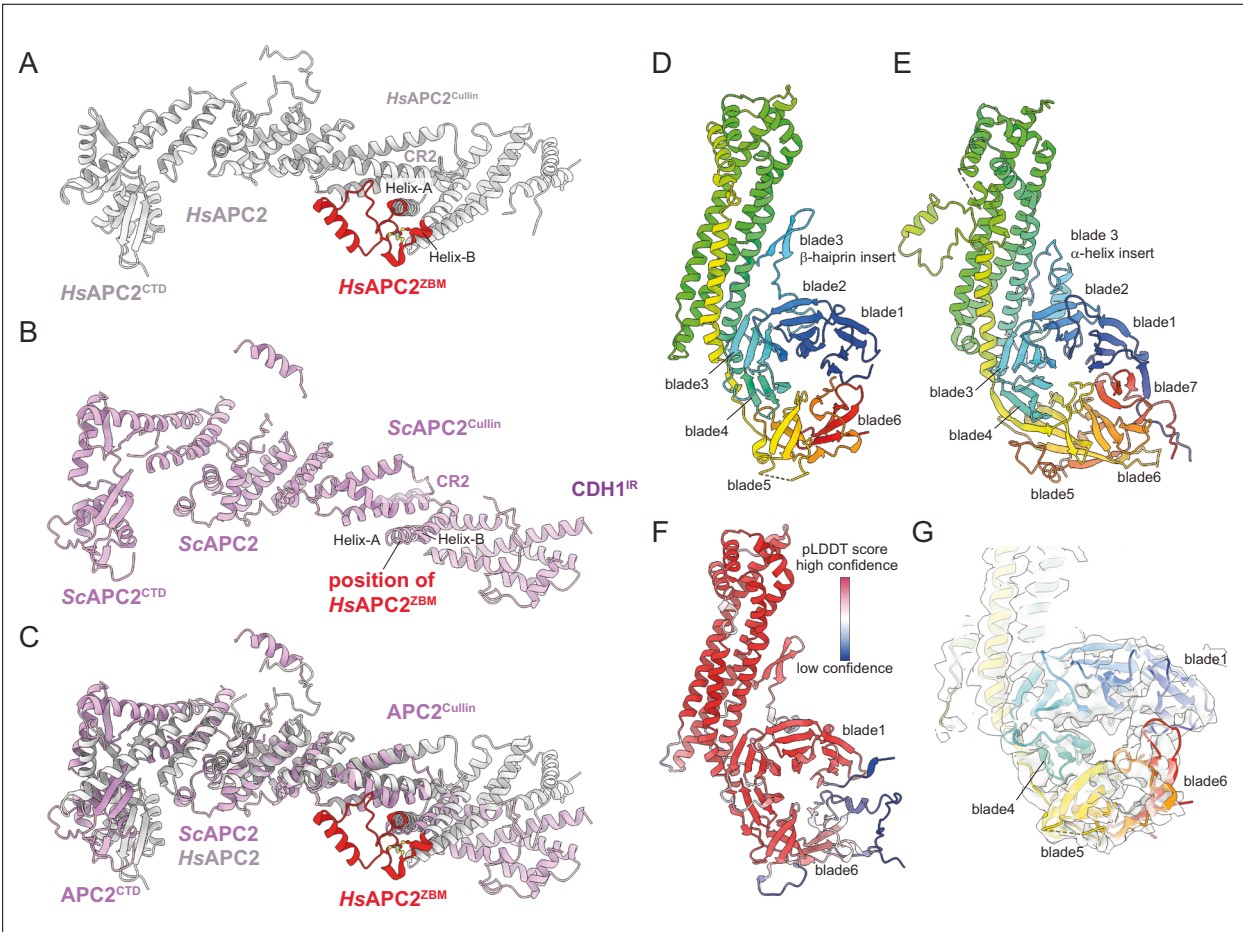

**Figure 5.** Comparison of *S. cerevisiae* and human APC2 and APC4 subunits. (**A**) Human APC2 with APC2ZBM of CRL2 coloured red. (**B**) *S. cerevisiae* APC2. The position of human APC2ZBM is indicated. (**C**) Superimposition *S. cerevisiae* APC2 (purple) with human APC2 (grey) on CRL2. Human APC2 from *Höfler et al., 2024* (PDB 7QE7). (**D**) *S. cerevisiae* APC4, colour-coded from N- to C-termini with a blue to red ramp. (**E**) Human APC4, colour-coded from N- to C-termini with a blue to red ramp. *S. cerevisiae* APC4WD40 is a six-bladed β-propeller in contrast to the seven bladed β-propeller of human APC4WD40. (**F**) AlphaFold2 prediction of *S. cerevisiae* APC4 colour-coded according to pLDDT score. No seventh bade is predicted. The α-helix in place of the seventh blade is predicted with low confidence. (**G**) Fit of *S. cerevisiae* APC4WD40 into the APC/CCDH1:Hsl1 cryo-EM map showing lack of density for a seventh blade. Human APC4 from *Höfler et al., 2024* (PDB 7QE7).

## CDC26/APC12

The N-terminal 13 residues of CDC26/APC12 insert into the TPR groove of APC6, adopting an extended conformation and mode of binding highly conserved with previous cryo-EM and crystallographic structures of human and *S. pombe* orthologs (*Chang et al., 2014*; *Chang et al., 2015*; *Wang et al., 2009*; *Zhang et al., 2010a*; *Figure 4*), although at the resolution of our APC/C$^{CDH1:Hsl1}$ cryo-EM map we cannot define an N-terminal acetyl group visualised in the crystal structure of the *S. pombe* APC6-APC12 complex (*Zhang et al., 2010a*). CDC26/APC12 then folds into an α-helix that packs against α-helices of the inner groove of the APC6 TPR super-helix, similar to the human and *S. pombe* APC6:APC12 complexes, as is the disordered C-terminus of CDC26/APC12. In neither the human and *S. cerevisiae* APC/C complexes, apart from APC6, do we observe interactions of CDC26/APC12 with other APC/C subunits. Thus, the role of CDC26/APC12 in stabilising the association of APC3, APC6 and APC9 with the APC/C complex (*Schwickart et al., 2004*; *Zachariae et al., 1998b*) is likely through its specific role in stabilising the APC6 TPR super-helix (*Wang et al., 2009*) which bridges APC3 with APC8 (*Figure 4E and F*).

## SWM1/APC13

In human APC/C, the APC13 N-terminus inserts into the TPR groove formed by the N-terminal TPR helix of APC8A (*Brown et al., 2015*; *Chang et al., 2014*; *Chang et al., 2015*). The extended chain of APC13 then interacts with structurally equivalent symmetry-related shallow grooves present on the outer TPR-helical ridges of APC8B, APC6A, APC6B, and APC3A (sites 4–7, *Figure 4F*). These shallow grooves are lined with side chains of aromatic residues that engage hydrophobic residues of APC13 (*Chang et al., 2015*). This pattern of TPR subunit engagement by APC13 is continued by APC16 interactions with equivalent sites on APC3B, APC7A, and APC7B (sites 1–3, *Figure 4F*). An earlier electron microscopy analysis of *S. cerevisiae* APC/C using an APC13-green fluorescent protein fusion positioned APC13 close to the APC6-APC3 interface (*Schreiber et al., 2011*), correlating with the position of human APC13. In our structure of *S. cerevisiae* APC/C$^{CDH1:Hsl1}$, we built APC13 into discontinuous cryo-EM density that spans APC8 to APC3. Similar to human APC13, *S. cerevisiae* APC13 engages the structurally equivalent sites on APC8B, APC6A, APC6B, and APC3A (sites 4–7, *Figure 4E*), with its N-terminus also inserting into the TPR groove of APC8A.

Human APC13 and *S. cerevisiae* SWM1/APC13 perform comparable structural roles, contacting the successive TPR subunits of the TPR lobe. We proposed previously that these contacts function to order the stacking of TPR homo-dimers of the TPR groove by breaking the symmetry of inter-TPR dimer interfaces. Although human and *S. cerevisiae* APC13 share only low sequence similarity (*Supplementary file 1b*), their conserved structural roles presumably explains the ability of human APC13 to substitute for the function of *S. cerevisiae* SWM1/APC13 in vivo (*Penkner et al., 2005*; *Schwickart et al., 2004*).

## APC9 (human APC16 ortholog)

In human APC/C, APC16 is visible as a single 40-residue extended α-helix that lies along the shallow grooves on the outer TPR-helical ridges of APC3B, APC7A and APC7B (sites 1-3, *Figure 4F*), forming equivalent interactions as observed for APC13 contacts with APC3, APC6 and APC8 (*Brown et al., 2015*; *Chang et al., 2014*; *Chang et al., 2015*). *S. cerevisiae* APC9 is over twice the size of APC16 with the two proteins sharing no clear sequence similarity. Based on a subunit deletion approach, we had previously assigned a region of APC9 to the C-terminal TPR super-helix of APC3A (*Schreiber et al., 2011*). This clearly differs from the association of APC16 with human APC/C. Guided by an AlphaFold2 prediction of APC9 (*Tunyasuvunakool et al., 2021*) and a complex of APC9 and APC3 (*Jumper et al., 2021*), we built 131 of 265 residues into the *S. cerevisiae* APC/C$^{CDH1:Hsl1}$ cryo-EM map as three discontinuous segments (*Figure 4C* and *Supplementary file 1b*). The C-terminal segment (segment 3) is a short three-turn α-helix that is structurally homologous to the mode of human APC16 binding to APC3B (*Figure 4E and F*). Segment 2, composed of three separated α-helices, docks onto the outer surface of APC3A, bridging the two turns of the TPR helix (*Figure 4E*). Finally, segment 1 bridges APC3A and APC6A (*Figure 4C* **insert**), likely explaining why APC9 deletion causes loss of both APC3 and APC9 from purified APC/C complexes (*Passmore et al., 2003*; *Zachariae et al., 1998b*).

## MND2/APC15

In human APC/C, the visible regions of APC15 adopt a relatively simple elongated structure, with the N-terminus of the protein inserting as an extended chain into the groove formed by the N-terminal TPR helix of APC5, then forming an α-helix (APC15[NTH]) that contacts the N-terminal α-helical domain of APC5, before engaging a groove at the interface of APC6A and APC8A. APC15 interacts with APC8A at a site structurally equivalent to the sites on APC8B, APC6 and APC3 engaged by the other small subunits APC9/APC16, APC12, and APC13 (site 8, *Figure 4B, D and F*). *S. cerevisiae* MND2/ APC15 also contacts the N-terminal α-helical domain of APC5, and APC6A and APC8A, as for human APC15, but has an overall more irregular structure, with its N-terminus docking into a groove formed between the APC5 TPR helix and the four-helical bundle of APC4 (*Figure 4A, C and E*). In contrast to human APC5, the APC5 TPR groove is occupied by a loop emanating from the APC5 N-terminal α-helical domain. An insert of *S. cerevisiae* APC15, not conserved in human, protrudes into the APC/C cavity, engaging the outer TPR helix of APC8B that forms the C-box binding site (*Figure 4A*). A stretch of ~40 acidic residues, common to both human and *S. cerevisiae* APC15, is C-terminal to the ordered regions of APC15 in both complexes.

In the human APC/C[MCC] structure, the mitotic checkpoint complex (MCC) was observed to adopt both closed and open conformations (APC/C[MCC]-closed and APC/C[MCC]-open), with APC/C[MCC]-closed being associated with disorder of APC15[NTH] and concomitant conformational changes of APC4 and APC5, whereas an ordered APC15[NTH] is associated with APC/C[MCC]-open (*Alfieri et al., 2016*; *Yamaguchi et al., 2016*). In APC/C[MCC]-closed, the UbcH10-binding site on the APC2:APC11 catalytic module is sterically occluded by the MCC. These structural observations explained why APC15 deletion in human APC/C abolished CDC20[MCC] auto-ubiquitylation in the context of APC/C[MCC] (*Alfieri et al., 2016*; *Uzunova et al., 2012*; *Yamaguchi et al., 2016*). Our observation that APC15[NTH] is structurally conserved in *S. cerevisiae* APC/C (Figure 4E) is consistent with a defect in CDC20[MCC] ubiquitylation caused by deleting APC15 (*Foster and Morgan, 2012*). In contrast to MND2/APC15 promoting CDC20[MCC] ubiquitylation during the SAC, in *S. cerevisiae*, MND2 specifically suppresses securin/Pds1 ubiquitylation by the Ama1 coactivator-APC/C complex that prevents premature sister chromatid segregation during meiosis (*Oelschlaegel et al., 2005*). The 14 sites of MND2 phosphorylation required for efficient APC/C[Ama1]-regulated progression through meiosis I (*Torres and Borchers, 2007*), are located within, and C-terminal to, the acidic stretch in MND2, and are therefore not visible in our structure.

A common feature of all the small APC/C subunits is that they form extended, mainly irregular structures, that simultaneously contact multiple large globular APC/C subunits. CDC26/APC12, SWM1/APC13 and human APC15 are anchored at their N-termini by inserting into the TPR grooves of APC6, APC8A, and APC5, respectively (Figure 4A and B). These interactions serve to stabilise the TPR groove through formation of protein-protein interactions (*Wang et al., 2009*). In human APC/C, a 40-residue loop of APC1[WD40] inserts into the TPR groove of APC8B, although this structural feature is not conserved in *S. cerevisiae* APC/C.

## Interactions of CDH1 with APC/C and conserved regulation by phosphorylation

Coactivators interact with the APC/C through an N-terminal C-box motif (*Schwab et al., 2001*) and a C-terminal IR tail (*Passmore et al., 2003*; *Vodermaier et al., 2003*). Both motifs are highly evolutionarily conserved. Interestingly, although the seven-residue C-box and two-residue IR tail lack obvious sequence similarity, the C-box binding site within the TPR helix of APC8B is structurally conserved with the IR-tail binding site on APC3, with an Arg residue, common to both, anchoring the motifs to their respective binding sites (*Chang et al., 2015*). The docking of the coactivator C-box motif to this site on APC8B/CDC23 rationalises the temperature-sensitive cell cycle mutations of *S. cerevisiae* CDC23 (*Sikorski et al., 1990*), and prior mutagenesis data implicating Asn405 of APC8/CDC23 in CDH1 binding (*Matyskiela and Morgan, 2009*). Additionally, other segments within CDH1[NTD] mediate APC/C-coactivator interactions. When in complex with the APC/C, CDH1[NTD] folds into five α-helices (labelled α2-α6; *Figure 6*). Four of these α-helices mediate APC/C – CDH1 interactions, three of which; α2, α3 and α6, that form anti-parallel interactions with APC1[PC] α-helices, being conserved with human CDH1 (*Figure 6B and E*). Unique to *S. cerevisiae* CDH1 is a three-turn α-helix (α5) that docks onto the outer TPR ridge of APC6B, running antiparallel to exposed α-helices, and bridging

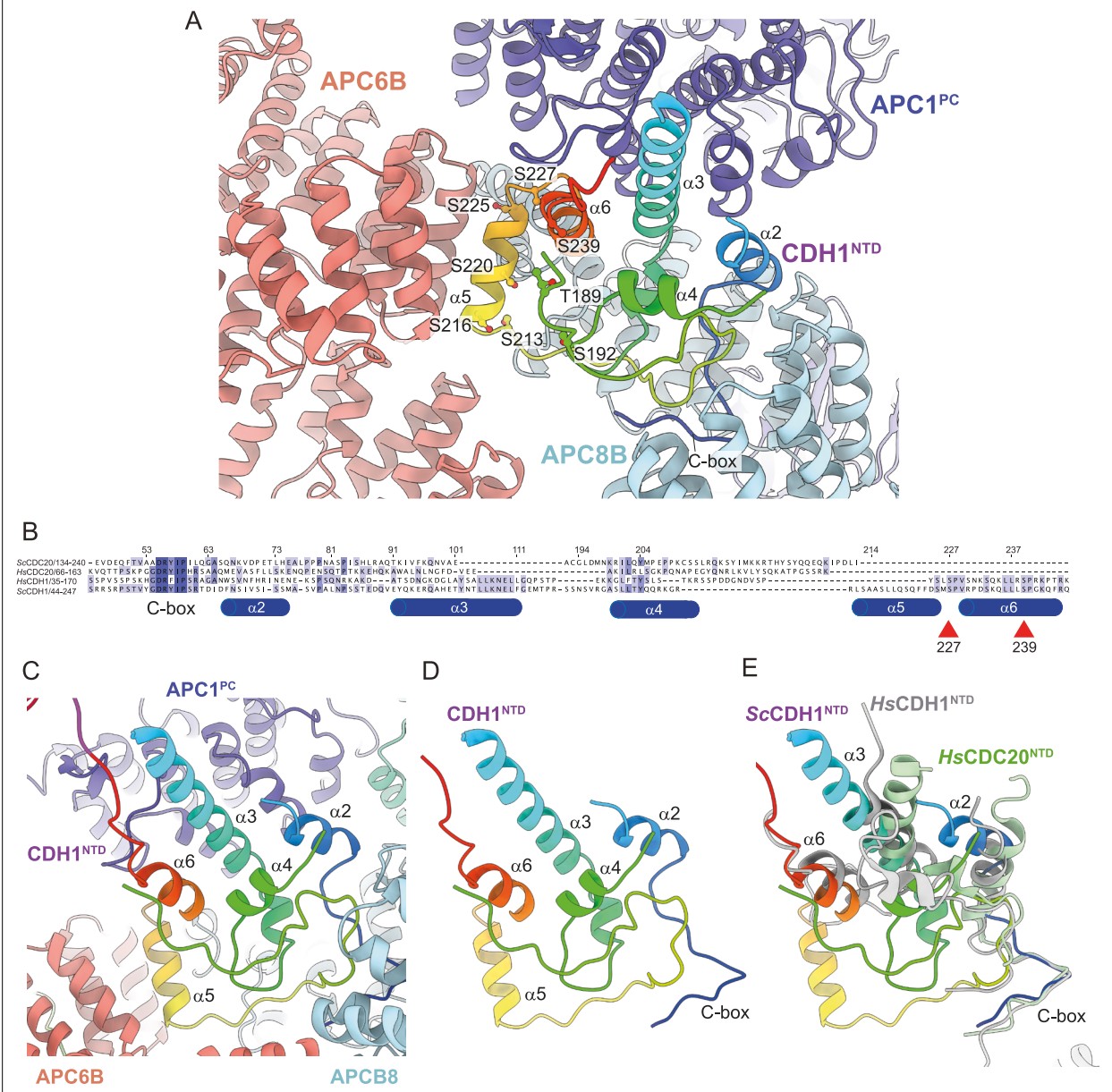

**Figure 6.** CDH1^NTD contacts to APC/C and control by phosphorylation. (**A**) In *S. cerevisiae* CDH1, α2, α3 and α6 contact the PC domain of APC1 (APC1^PC) (conserved with human CDH1). α2 also contacts APC8B. α5, unique to *S. cerevisiae* CDH1, forms extensive contacts to APC6B. Human CDC20^NTD (light green) shares α2 and α3 with CDH1^NTD. (**B**) MSA of *S. cerevisiae* and human CDH1^NTD and CDC20^NTD showing α-helices α2 to α6 of *S. cerevisiae* CDH1^NTD and the conserved sites of CDK phosphorylation on CDH1 (red arrows). (**C**) *S. cerevisiae* CDH1^NTD colour-coded from N- to C-termini with a blue to red ramp interacting with APC1^PC, APC6B and APC8B. (**D**) Same view as (**c**) without APC1^PC, APC6B and APC8B. (**E**) Superimposition of *S. cerevisiae* CDH1^NTD coloured blue-to-red, human CDH1^NTD (grey) (from ***Höfler et al., 2024***, PDB 7QE7) and human CDC20^NTD (light green) (from ***Zhang et al., 2019*** PDB 6Q6G). α-helices α2, α3 and α6 are conserved in all three structures.

APC8B (***Figure 6A and C***). CDK phosphorylation of CDH1^NTD inhibits CDH1 binding to the APC/C, and substituting alanines for all eleven predicted CDK sites in *S. cerevisiae* CDH1 almost completely eliminated CDH1 phosphorylation in mitosis (***Zachariae et al., 1998a***). Mimicking six CDK phosphosites with aspartates abolished binding of CDH1 to the APC/C in vivo, whereas a mutant with non-phosphorylatable sites bound APC/C more tightly than wild type CDH1 (***Höckner et al., 2016***). Five of these CDK sites, with additional non-consensus CDK sites, were reported to be phosphorylated in vivo (***Hall et al., 2004***). Mapping these phosphosites onto the *S. cerevisiae* APC/C^CDH1:Hsl1 structure suggested a rationale for how CDH1^NTD phosphorylation inhibits binding of CDH1 to the APC/C

(*Figure 6A*). Two CDK sites (Ser227, Ser239) are conserved in human CDH1 (*Figure 6B*; *Chang et al., 2015*), and map on or close to the α6 helix that docks onto APC1$^{PC}$ (*Figure 6A*). Phosphorylation of either site in human CDH1 contributed significantly to CDH1 inactivation (*Chang et al., 2015*), likely by disrupting CDH1$^{NTD}$-APC1$^{PC}$ interactions. Thus, this analysis reveals an evolutionarily conserved molecular mechanism for inhibiting CDH1 binding to the APC/C through CDK phosphorylation. Additionally, three reported non-consensus CDK-phosphosites within α5 (Ser216, Ser220, Ser225) (*Hall et al., 2004*) would likely interfere with α5 interactions with APC6B and APC8B (*Figure 6A*).

The recent 2.9 Å resolution cryo-EM structure of APC/C$^{CDH1:EMI1}$ revealed an N-terminal amphipathic α-helix of CDH1 (CDH1$^{α1}$) interacting through non-polar interactions with a hydrophobic groove in the APC1$^{WD40}$ domain (*Höfler et al., 2024*). This α-helix is conserved in metazoan coactivators, but consistent with MSA analysis, is not present in *S. cerevisiae* CDH1.

Finally, the mode of binding of CDH1$^{IR}$ to the IR-tail binding site on APC3A is conserved with that of human APC/C$^{CDH1:EMI1}$ (*Brown et al., 2015*; *Chang et al., 2015*; *Höfler et al., 2024*; *Matyskiela and Morgan, 2009*; *Figure 7A–D*). In *S. cerevisiae* APC/C, the interaction of APC9 with APC3A (segment 2 of APC9) and APC3B (segment 3 of APC9) partially mimics the respective APC7 and APC16 interactions with human APC3 (*Figure 7E and F*). APC9 segment 2 interactions with the C-terminal TPR helix of APC3A may stabilise the CDH1$^{IR}$ tail-binding site of APC3A.

## Comparison of apo-APC/C with APC/C$^{CDH1:Hsl1}$

Biochemical studies on both metazoan and *S. cerevisiae* APC/C revealed that coactivators function as substrate adaptors (*Burton and Solomon, 2001*; *Hilioti et al., 2001*; *Kraft et al., 2005*; *Schwab et al., 2001*), and also enhance APC/C catalytic activity (*Kimata et al., 2008*; *Van Voorhis and Morgan, 2014*). For *S. cerevisiae* APC/C, catalytic enhancement results from a substantially more efficient interaction of the APC/C with its E2s Ubc4 and Ubc1 (*Van Voorhis and Morgan, 2014*). Similarly, the affinity of the human APC/C for UbcH10 is increased three-fold when in complex with CDH1 (*Chang et al., 2014*). In human APC/C, the association of CDH1 induces a conformational change of the APC2:APC11 catalytic module from a downwards state, in which the UbcH10-binding site is blocked, to an upwards state competent to bind UbcH10 (*Chang et al., 2014*). Comparing unphosphorylated *S. cerevisiae* apo-APC/C and APC/C$^{CDH1:Hsl1}$ cryo-EM structures showed that in both states the APC2:APC11 catalytic module adopts the same raised position (*Figure 8A and B*), similar to human APC/C$^{CDH1:EMI1}$. Docking Ubc4 onto the RING domain of APC11 (based on the *S. cerevisiae* Not4:Ubc4 coordinates; PDB 5AIE *Bhaskar et al., 2015*) indicated that the conformation of the APC2:APC11 catalytic module in apo-APC/C does not sterically occlude Ubc4 binding (*Figure 8C and D*). Thus, the mechanism by which coactivator stimulates *S. cerevisiae* APC/C catalytic activity through enhancing E2 binding differs from human APC/C, although the structural mechanism is not defined in our study.

Why isn't *S. cerevisiae* apo-APC/C in an inactive conformation? The interaction of CDH1$^{NTD}$ with APC1 and APC8B of human APC/C explains how CDH1 association promotes a conformational change of the APC/C that results in the upward movement of the APC2:APC11 catalytic module (*Chang et al., 2014*; *Höfler et al., 2024*). In the human ternary complex, CDH1$^{NTD}$ disrupts the interaction between APC1$^{PC}$ and APC8B by binding to a site on APC1$^{PC}$ that overlaps the site in contact with APC8B. This wedges APC1$^{PC}$ and APC8B apart. The resultant downwards-displacement of APC8B tilts the platform to translate the APC2$^{CTD}$:APC11 module upwards. In contrast to human APC/C, in the more open structure of *S. cerevisiae* apo-APC/C, the TPR lobe does not directly contact APC1$^{PC}$, and thus the position of APC8B relative to the APC8B – CDH1$^{NTD1}$-binding site does not impede CDH1 association. Instead, on binding to *S. cerevisiae* APC/C, of CDH1$^{NTD}$ interconnects the TPR lobe with APC1$^{PC}$ to reinforce TPR and platform lobe interactions (*Figure 8A and B*; lower panels). To accommodate closure of the APC/C cavity, conformational changes are distributed over the subunits of the TPR and platform lobes, including a tilt of APC1$^{PC}$ relative to the combined domains of APC1$^{Mid}$–APC1$^{WD40}$ (*Figure 8—figure supplement 1* and *Video 1*).

## Mechanism of APC/C activation by phosphorylation

Hyperphosphorylation of APC/C subunits APC1 and APC3 is required for CDC20 to activate the APC/C (*Fujimitsu et al., 2016*; *Golan et al., 2002*; *Kraft et al., 2005*; *Kramer et al., 2000*; *Lahav-Baratz et al., 1995*; *Qiao et al., 2016*; *Rudner and Murray, 2000*; *Shteinberg et al., 1999*; *Zhang*

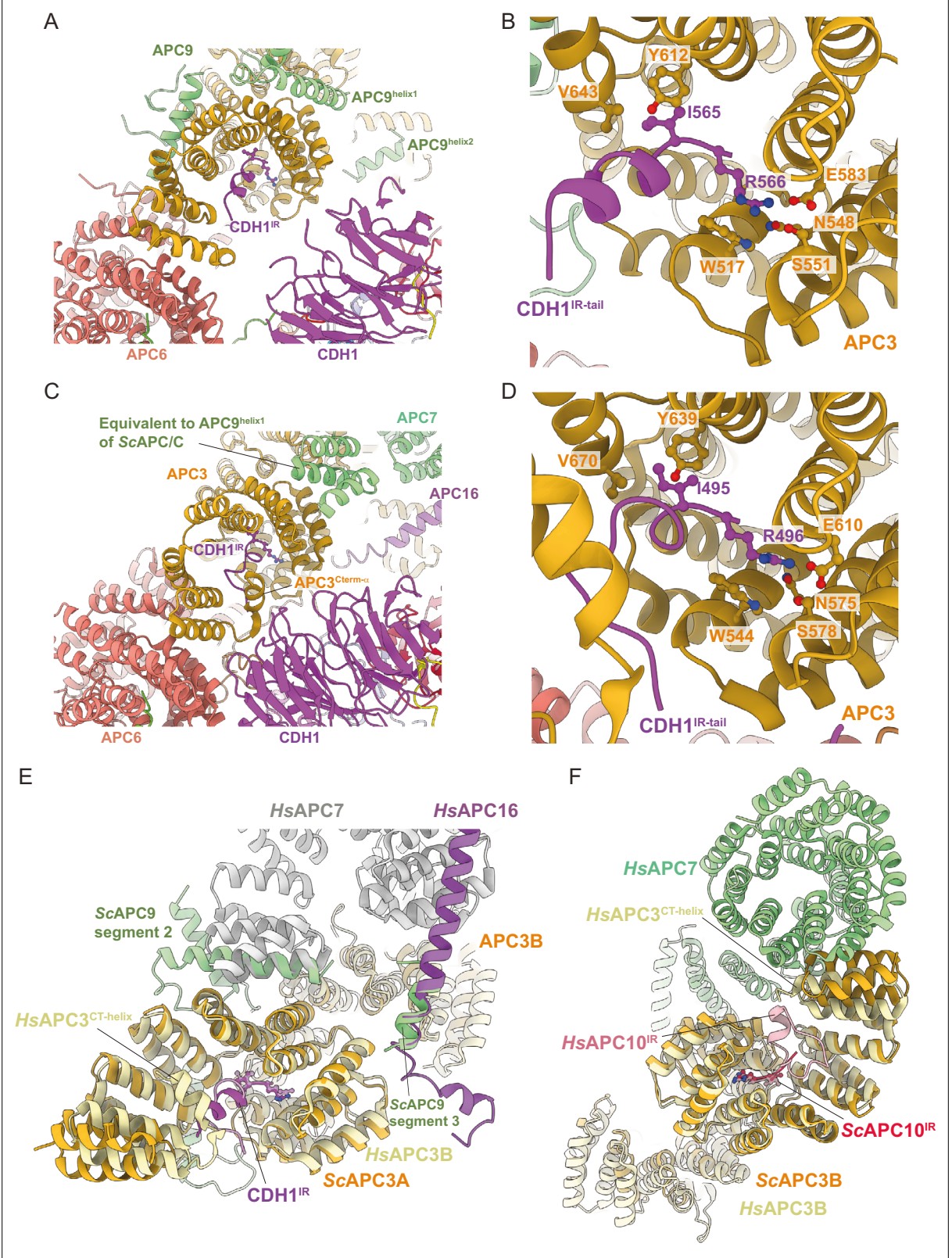

**Figure 7.** Comparison of CDH1[IR] binding to *S.cerevisiae* and human APC/C[CDH1] complexes. (**A**) Overall view of the CDH1[IR]-binding site on *S. cerevisiae* APC/C[CDH1:Hsl1]. (**B**) Zoomed view of this complex. (**C**) Overall view of the CDH1[IR]-binding site on human APC/C[CDH1:EMI1]. (**D**) Zoomed view of this complex. The structure of human APC/C[CDH1:EMI1] from *Höfler et al., 2024* (PDB 7QE7). (**E, F**) APC9 of *S. cerevisiae* APC/C partially mimics APC7 and APC16 of human APC/C. (**E**) In *S. cerevisiae* APC/C, the interaction of APC9 with APC3A (segment 2 of APC9) and APC3B (segment 3 of APC9) partially mimics

*Figure 7 continued on next page*

*Figure 7 continued*

the respective APC7 and APC16 interactions with human APC3. APC9 segment 2 interactions with the C-terminal TPR helix of APC3A may stabilise the CDH1$^{IR}$-binding site of APC3A. (**F**) Similarly, in human APC/C, the interface of APC7B with APC3B might function to stabilise the APC10$^{IR}$-binding site on APC3B. Human APC/C$^{CDH1:EMI1}$ from *Höfler et al., 2024* (PDB 7QE7).

*et al., 2016b*). In metazoan unphosphorylated apo-APC/C, an auto-inhibitory (AI) segment incorporated into the 300s loop inserted within the WD40 domain of APC1 (APC1$^{300L}$) sterically occludes the C-box binding site on APC8B (APC8$^{C-box}$) thereby competing for CDC20 binding (*Fujimitsu et al., 2016*; *Qiao et al., 2016*; *Zhang et al., 2016b*). Phosphorylation of residues within (APC1$^{300L}$), including the AI segment, displaces the AI segment from APC8$^{C-box}$ allowing CDC20 binding. Multiple lines of data support this model, including the observation of cryo-EM density for the AI segment bound to APC8$^{C-box}$ (*Höfler et al., 2024*; *Zhang et al., 2016b*), the constitutive activation of APC/C$^{CDC20}$ on deleting APC1$^{300L}$, and the location of multiple CDK and PLK1 sites within APC1$^{300L}$ (*Fujimitsu et al., 2016*; *Höfler et al., 2024*; *Li et al., 2016*; *Qiao et al., 2016*; *Zhang et al., 2016b*). Replacing these phosphosites with glutamates activated APC/C$^{CDC20}$, whereas in contrast, mutations to Ala prevented CDK activation of APC/C$^{CDC20}$ (*Fujimitsu et al., 2016*; *Höfler et al., 2024*; *Qiao et al., 2016*; *Zhang et al., 2016b*). Our cryo-EM structure suggests that activation of *S. cerevisiae* APC/C$^{CDC20}$ by phosphorylation operates by a different mechanism. We observe no cryo-EM density occupying the C-box binding site of APC8B in unphosphorylated apo-APC/C (*Figure 9A and B*). Arguably, the low resolution of this structure means that we cannot definitively state that APC8$^{C-box}$ is not occluded in the unphosphorylated apo-state. However, the Morgan lab recently reported that deleting the APC1 WD40 loop of *S. cerevisiae* (residues 225-365), equivalent to human APC1$^{300L}$, caused only a minor growth defect, but importantly no evidence for APC/C hyperactivation (*Ng et al., 2024*), a finding that is not consistent with a model in which *S. cerevisiae* and human APC/C share a common phospho-regulatory mechanism.

Superimposing both apo-APC/C states (phosphorylated and unphosphorylated) reveals that the only obvious difference between the two is a small relative shift of the TPR lobe (*Figure 9C–E*). Since APC/C phosphorylation stimulates CDC20 association, and not CDH1, we would expect specific and perhaps localised conformational changes in the APC/C. It is possible that in contrast to the activation of human APC/C$^{CDC20}$ through release of a negative auto-inhibitory segment, phosphorylated regions of *S. cerevisiae* APC/C might contact CDC20 directly or indirectly to enhance its binding. We note that 'unphosphorylated' APC/C isolated from insect cells is partially phosphorylated prior to its treatment with CDK2 (*Supplementary file 1a*). Although CDK2-dependent phosphorylation causes the expected mobility shift in APC3/CDC27 (*Figure 1—figure supplement 1A*), and an increase in phosphorylation sites (*Supplementary file 1a*), it is possible that 'unphosphorylated' APC/C may already contain activating phosphorylation sites, thus explaining the absence of conformational change on in vitro CDK2-mediated phosphorylation. Our phosphoproteomic study did not assess phospho-site stoichiometry.

## Discussion

We present a comparative study of human and *S. cerevisiae* APC/C. This revealed a similar overall architecture, with the largest difference being the absence of an APC7 homolog in *S. cerevisiae*. A recent study discovered a specific role for APC7 in mammalian brain development (*Ferguson et al., 2022*). Homozygous ANAPC7 mutations in humans, leading to loss of APC7 protein, underlies an inherited intellectual disability syndrome, caused by defective APC/C$^{ΔAPC7}$-mediated degradation of the chromosome-associated protein Ki-67 (*Ferguson et al., 2022*). Thus in mammals, APC7 confers specific post-mitotic functions in the developing brain through selective protein-recognition. The mode of binding of the CDH1 coactivator subunit is also very similar, except for the presence in human APC/C of the N-terminal α-helix (CDH1$^{α1}$), not conserved in *S. cerevisiae* APC/C, that mediates contacts to APC1$^{WD40}$. Also conserved is the mechanism of inactivation of CDH1 by CDK phosphorylation of two conserved serine residues on CDH1 at the CDH1$^{NTD}$ – APC1$^{PC}$ interface. Major differences between human and *S. cerevisiae* APC/C are how coactivators enhance APC/C affinity for E2s (UbcH10 in human *Chang et al., 2014*, Ubc1 and Ubc4 in *S. cerevisiae Van Voorhis and Morgan, 2014*), and how APC/C$^{CDC20}$ is activated by CDK-mediated APC/C phosphorylation. What is clear

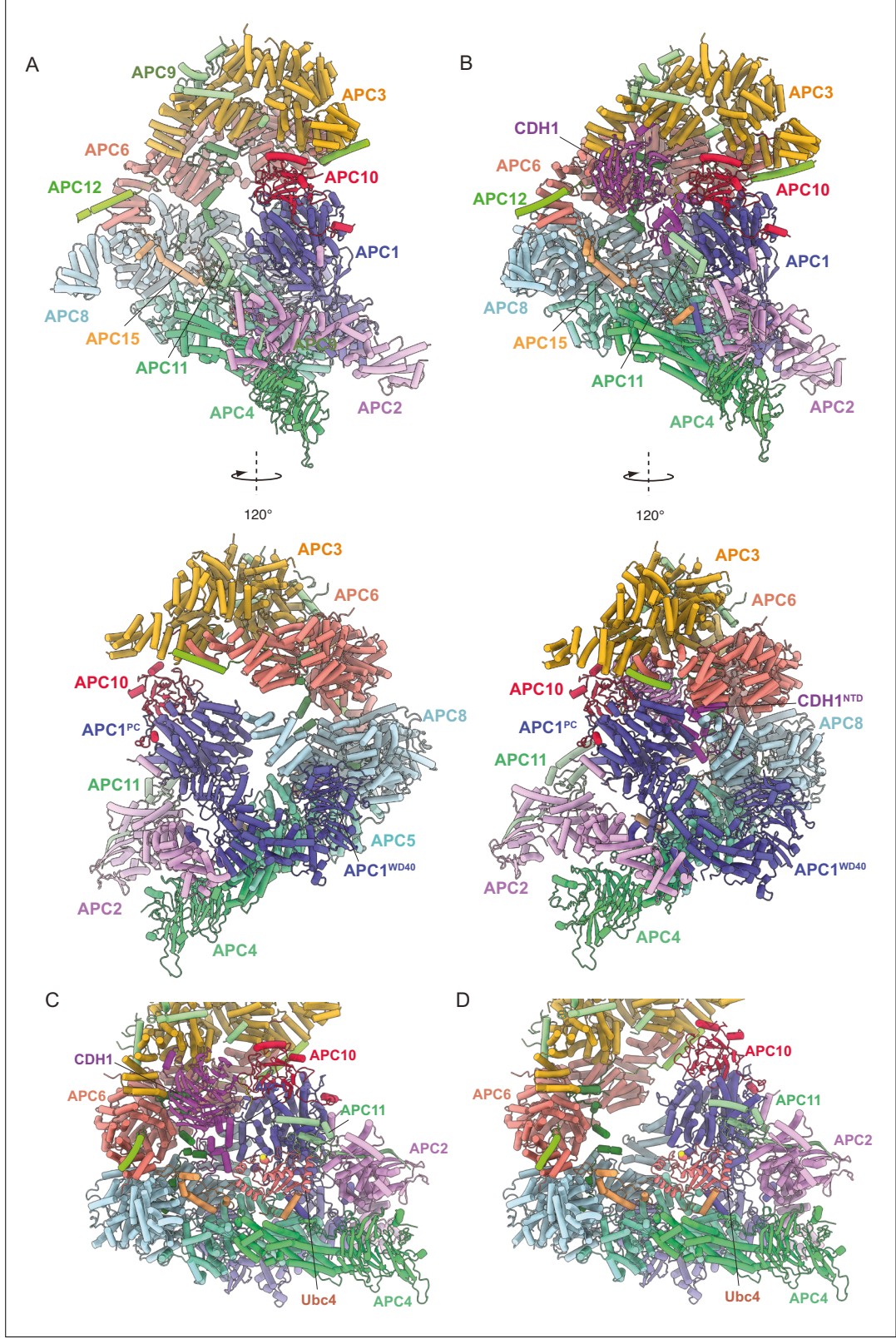

**Figure 8.** Comparing apo-APC/C and the APC/C$^{CDH1:Hsl1}$ complex shows both apo-APC/C and the APC/C$^{CDH1:Hsl1}$ complexes are competent to bind the Ubc4. (**A**) Two views of apo-APC/C. (**B**) Two views of APC/C$^{CDH1:Hsl1}$. The coordinates were superimposed on APC1$^{PC}$. In the apo-APC/C complex, there are no contacts between APC1$^{PC}$ of the platform module and APC6B and APC8B of the TPR lobe. In the APC/C$^{CDH1:Hsl1}$, CDH1$^{NTD}$ bridges APC1$^{PC}$ with

*Figure 8 continued on next page*

*Figure 8 continued*

APC6B and APC8B. The overall conformations are similar, specifically the APC2:APC11 catalytic module adopts a raised conformation in both states. (**C**) Model of APC/C$^{CDH1:Hsl1}$ in complex with Ubc4. (**D**) Model of apo-APC/C in complex with Ubc4. The position of the APC2:APC11 catalytic module in apo-APC/C does not exclude Ubc4 binding. Model of APC11:E2 based on the *S. cerevisiae* Not4 RING:Ubc4 complex (PDB 5AIE; *Bhaskar et al., 2015*).

The online version of this article includes the following figure supplement(s) for figure 8:

**Figure supplement 1.** Conformational change of APC1 on conversion from apo-APC/C to APC/C$^{CDH1:Hsl1}$.

---

from our structural analysis is that unlike human APC/C, in the apo-state of *S. cerevisiae* APC/C, the E2-binding site on APC11$^{RING}$ is accessible, and that CDH1 does not promote a substantial conformational change of the APC2:APC11 catalytic module. However, how CDH1 enhances E2 affinity for APC/C isn't clear from a comparison of the apo-APC/C and APC/C$^{CDH1:Hsl1}$ ternary structures. It is likely that the structures of APC/C$^{CDH1:Hsl1}$ in complex with E2s would reveal this mechanism. Finally, the mechanism of stimulating CDC20-binding to the APC/C through phosphorylation-dependent relief of an auto-inhibitory segment of APC1, as in human APC/C (*Fujimitsu et al., 2016*; *Qiao et al., 2016*; *Zhang et al., 2016b*), is not conserved in *S. cerevisiae* APC/C, a structural finding that is consistent with in vivo data (*Ng et al., 2024*). Since comparison of all three reported structures did not reveal a likely mechanism, one explanation could be that a phosphorylated region of the APC/C either directly or indirectly contacts CDC20 to enhance its affinity for the APC/C.

# Materials and methods
## Cloning and expression of recombinant *S. cerevisiae* APC/C and CDH1:Hsl1

*S. cerevisiae* APC/C genes were previously cloned into a modified MultiBac system (*Zhang et al., 2016c*). Coding sequences for CDH1 and Hsl1$^{667-872, 1k}$ (Hsl1$^{667-872, K672R/ K683R/ K701R/ K701R/ K710R/ K747R/ K760R/ K785R/ K788R/ K796R/ K812R/ K825R/ K827R/ K840R/ K843R/ K851R/ K852R/ K860R/ K861R/ K869R}$) were cloned into a pU1 vector by USER methodology (*Zhang et al., 2016c*).

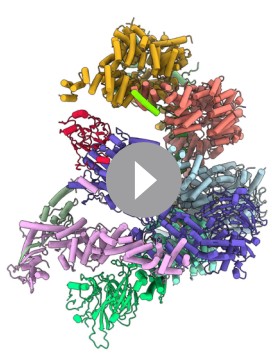

**Video 1.** CDH1-induces conformational changes in APC/C. The video shows a morph of apo-APC/C to the ternary APC/C$^{CDH1:Hsl1}$ complex indicating the conformational changes induced in APC/C as a consequence of CDH1 binding. The two conformational states were superimposed on the N-terminus of APC1. On transition to the ternary APC/C$^{CDH1:Hsl1}$ state, APC3 and APC6 move closer to the APC1$^{PC}$ domain, as a result of the CDH1 N-terminal domain binding at the interface of APC1$^{PC}$ and APC6. The catalytic module of APC2-APC11 does not change conformation.
https://elifesciences.org/articles/100821/figures#video1

For APC/C expression, High-Five cells (Invitrogen) at a density of $2 \times 10^6$ cells/mL were co-infected with two (apo-APC/C) or three (APC/C$^{CDH1:Hsl1}$) pre-cultures of High-Five cells each pre-infected with one of the recombinant APC/C baculoviruses. APC/C expression was performed for 48 hr. Cells were harvested by centrifugation at $1000 \times g$ for 10 min at 4 °C.

## Purification of recombinant *S. cerevisiae* APC/C

All purification steps were carried out at 4 °C and are the same for the apo-APC/C and the ternary APC/C$^{CDH1:Hsl1}$ complex. The pellets from insect cells were resuspended in lysis buffer [50 mM Tris-HCl (pH 8.3), 200 mM NaCl, 5% glycerol, 2 mM DTT, 1 mM EDTA, 2 mM benzamidine, 0.5 mM PMSF, 5 units/mL benzonase (Sigma-Aldrich) and Complete EDTA-free protease inhibitor (Roche)]. After sonication, the cell lysate was centrifuged for 60 min at $48,000 \times g$ and filtered (0.45 μm). The cleared supernatant was loaded onto a 5 mL Strep-Tactin Superflow Cartridge (QIAGEN) at 1 mL/min. The column was washed with APC/C buffer [50 mM Tris-HCl (pH 8.0),

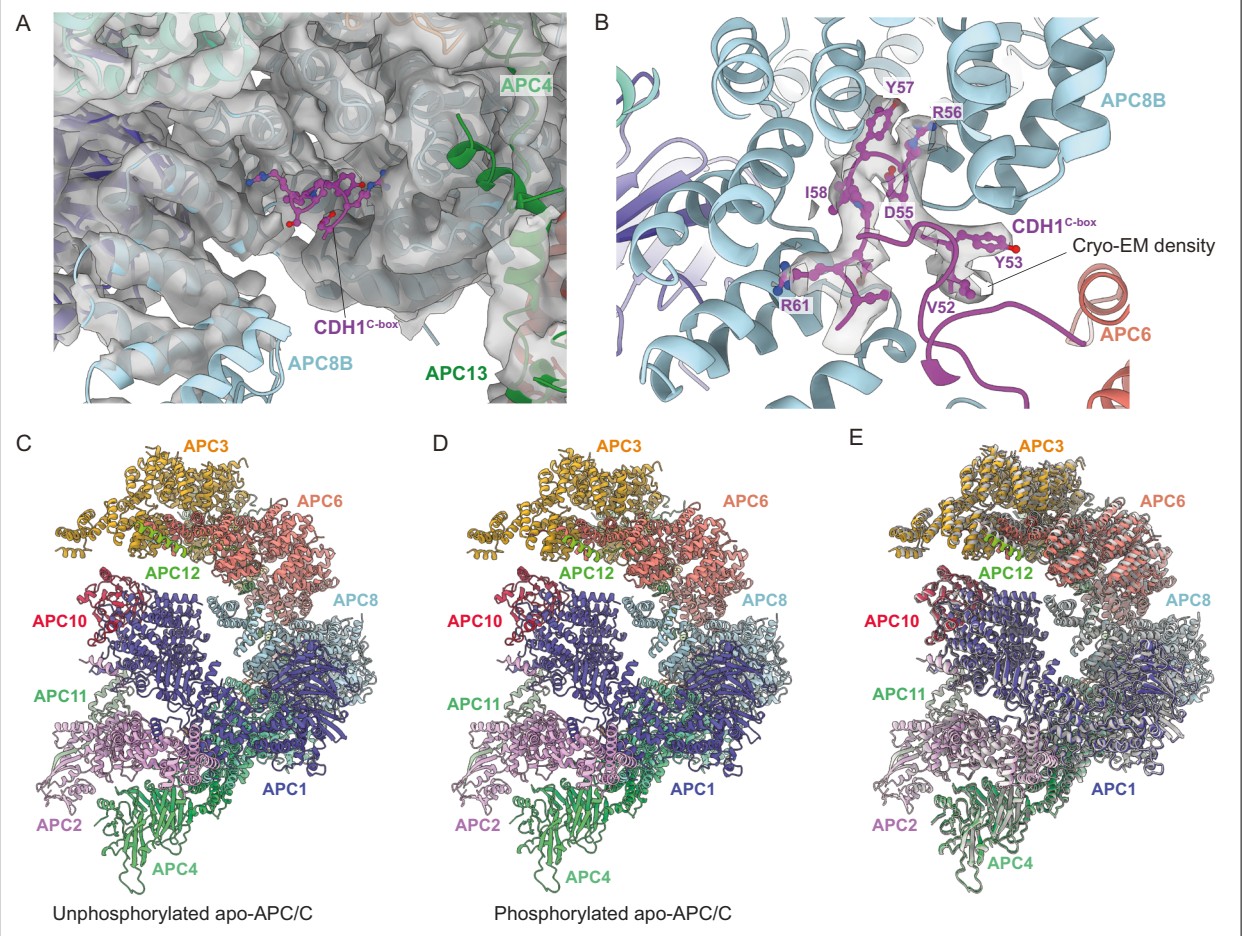

**Figure 9.** Regulation of APC/C by phosphorylation. The C-box binding site is un-obstructed in unphosphorylated apo-APC/C. (**A**) Cryo-EM density from the unphosphorylated apo-APC/C map corresponding to the C-box binding site of APC8B with the fitted CDH1$^{C-box}$ modelled from APC/C$^{CDH1:Hsl1}$. We observe no density at the APC8B C-box binding site. (**B**) Cryo-EM map density for the CDH1$^{C-box}$ from the APC/C$^{CDH1:Hsl1}$ reconstruction with fitted coordinates. (**C–E**) Comparison of unphosphorylated apo-APC/C and phosphorylated apo-APC/C: (**C**) Unphosphorylated apo-APC/C. (**D**) Phosphorylated apo-APC/C. (**E**) Superimposition of unphosphorylated apo-APC/C (coloured by subunit assignment) and phosphorylated apo-APC/C (grey). There is a small relative shift of the TPR lobe. The coordinates were superimposed on APC1$^{PC}$.

200 mM NaCl, 5% (v/v) glycerol, 2 mM DTT, 1 mM EDTA and 2 mM benzamidine]. The recombinant APC/C complex was eluted with the wash buffer supplemented with 2.5 mM desthiobiotin (Sigma-Aldrich). Before loading the elution fractions onto a 6 mL Resource Q anion-exchange column (Cytiva), they were diluted 1.6-fold into buffer A without NaCl [buffer A: 20 mM HEPES (pH 8.0), 120 mM NaCl, 5% (v/v) glycerol and 2 mM DTT]. The Resource Q column was washed with buffer A and the APC/C was eluted with a gradient of buffer B [20 mM HEPES (pH 8.0), 1 M NaCl, 5% (v/v) glycerol and 2 mM DTT]. The resulting elution was concentrated to around 3 mg/mL and ultracentrifuged for 10 min at 40,000 × *g*. The soluble supernatant was loaded onto a Superose 6 Increase 3.2/300 column (Cytiva) equilibrated in APC/C gel-filtration buffer [20 mM HEPES (pH 8.0), 150 mM NaCl and 0.5 mM TCEP]. The gel filtration was run on a ÄKTAmicro (Cytiva) with a flow rate of 50 µL/min.

## In vitro phosphorylation of recombinant *S. cerevisiae* APC/C

Resource Q apo-APC/C peak fractions were diluted twofold into buffer A without NaCl and concentrated. The concentrated sample was treated with CDK2–cyclin A3–Cks2 (*Zhang et al., 2016b*) in a molar ratio of 1:1.5 (APC/C: kinases) in a reaction buffer of 40 mM HEPES (pH 8.0), 10 mM MgCl$_2$, 5 mM ATP, 50 mM NaF, 0.1 µM okadaic acid and 20 mM β-glycerophosphate. The reaction mixture was incubated at 30 °C for 30 min. Subsequently, the phosphorylated protein was run on a Superose Increase 6 3.2/300 column (Cytiva) with APC/C gel-filtration buffer [20 mM HEPES (pH 8.0), 150 mM

NaCl, 0.5 mM TCEP]. For the ubiquitination assays, the reaction was stopped with 15 µM CDK1/2 Inhibitor III (Calbiochem) without further purification steps.

## Ubiquitination assays

APC/C ubiquitination assays were adopted from *Passmore et al., 2005*. $^{35}$S-labelled substrates (*S. cerevisiae* Hsl1) and unlabelled *S. cerevisiae* wildtype CDH1 were prepared in vitro using the TNT T7 Quick Coupled Transcription/Translation System (Promega). Each reaction contained 60 nM recombinant *S. cerevisiae* APC/C, 0.5 µL of $^{35}$S-labelled substrate and 2 µL CDH1 in a 10 µL reaction volume with 40 mM Tris.HCl (pH 7.5), 10 mM $MgCl_2$, 0.6 mM DTT, 5 mM ATP, 150 nM UBA1, 300 nM Ubc4, 70 µM ubiquitin, 200 ng ubiquitin aldehyde (Enzo), and 2 µM LLnL (N-Acetyl-Leu-Leu-Norleu-aldehyde) (Sigma-Aldrich). Reactions were incubated at 30 °C for 2 hr and terminated adding SDS/PAGE loading buffer. Samples were analysed by NuPAGE 4–12% Bis-Tris protein gels (ThermoFisher Scientific). Gels were fixed and stained with InstantBlue (Expedeon) followed by drying and exposure to X-ray Film (Photon Imaging Systems).

## Electron microscopy

Freshly purified APC/C samples were visualised by negative-staining EM to assess the quality and homogeneity of the sample. Micrographs were recorded on a CryoSpirit electron microscope (Thermo Fisher Scientific) at an accelerating voltage of 120 keV and at a defocus of approximately –1.5 µm. For cryo-EM, Quantifoil R3.5/1 grids coated with a layer of continuous carbon film (approximately 50 Å thick) were glow-discharged for 30 s before deposition of 3 µL fresh sample (~0.12 mg/mL), blotted for 8 s, and vitrified by plunging into liquid ethane with a custom-made manual plunger at 4 °C. Micrographs were collected on a Titan Krios microscope (Thermo Fisher Scientific) at an acceleration voltage of 300 keV and Falcon III direct electron detector. Micrographs were taken using EPU software (Thermo Fisher Scientific) at a nominal magnification of 59,000 x, with a calibrated pixel size of 1.38 Å. Fifty nine movie frames with an average electron dose of 59 e$^-$/Å$^2$ were recorded in integration mode for 1.49 s. The defocus range was set at –2.0 to –4.0 µm.

## Image processing

All movie frames were aligned and averaged with MotionCor2 (*Zheng et al., 2017*) or RELION's own implementation of the MotionCor2 algorithm (*Zivanov et al., 2018*). Contrast transfer function parameters were calculated with Gctf (*Zhang, 2016a*). Particles were automatically selected by template-free particle picking in Gautomatch (developed by Kai Zhang, http://www.mrc-lmb.cam.ac.uk/kzhang/Gautomatch/). All subsequent steps were performed in RELION-2/3 (*Zivanov et al., 2018*). Sorting of particle images was performed by 2D and 3D classification, using 60 Å low-pass filtered human unphosphorylated apo-APC/C EM map (EMD-3386) (*Zhang et al., 2016b*) as an initial 3D reference. The reconstruction generated from all the corresponding *S. cerevisiae* particles, low-pass filtered at 40 Å, was used as a subsequent reference. Finally, the data set was subject to either particle polishing or Bayesian particle polishing (*Zivanov et al., 2018*), CTF refinement and an extra 3D classification step to discard remaining bad particles. All resolution estimations were based on the gold-standard Fourier shell correlation (FSC) calculations using the FSC = 0.143 criterion. *Table 1* summarises EM reconstructions obtained in this work.

To improve map resolution, multi-body refinement was performed in in RELION 3.0 (*Nakane et al., 2018*). For the APC/C$^{CDH1:Hsl1}$ reconstruction, three masks were generated: mask 1 encompassed the TPR lobe (APC3, APC6, APC8, APC9, APC13); mask 2 included the platform (APC1, APC4, APC5, APC10, APC15 and CDH1 subunits and Hsl1), mask 3 surrounded the APC2:APC11 catalytic module. The resultant maps were determined at 4.1 Å, 4.06 Å, and 7.36 Å, respectively. For the apo-APC/C and phosphorylated apo-APC/C, two masks were used: mask1 contained the TPR lobe subunits; mask 2 included the platform and catalytic subunits. The apo-APC/C maps achieved resolutions of 4.9 Å and 4.58 Å, respectively, whereas the phosphorylated apo-APC/C maps have resolutions of 4.18 Å and 4.23 Å, respectively. The maps obtained from multi-body refinement demonstrated significantly improved definition of cryo-EM densities which facilitated model building.

## Model building of APC/C

Model building of the apo-APC/C, APC/C<sup>CDH1:Hsl1</sup> structures and phosphorylated apo-APC/C were performed in COOT (*Emsley and Cowtan, 2004*). Initially, available atomic structures of human unphosphorylated apo-APC/C (5G05) (*Zhang et al., 2016c*) and human APC/C<sup>CDH1:EMI1</sup> (4UI9) (*Chang et al., 2015*) and prior structures of human (*Chang et al., 2015*; *Höfler et al., 2024*), *E. cuniculi* (*Zhang et al., 2010b*), *S. cerevisiae* (*Au et al., 2002*), and *S. pombe* (*Zhang et al., 2013a*; *Zhang et al., 2010a*) APC/C subunits were rigid-body fitted as individual subunits into the cryo-EM maps in Chimera (*Yang et al., 2012*). Subsequently, the different human and *S. cerevisiae* APC/C protein subunit sequences were aligned using the MUSCLE program (*Edgar, 2004*) and Jalview (*Waterhouse et al., 2009*). Based on the multiple alignment results, the amino acids from the human fitted atomic structure were substituted for the corresponding *S. cerevisiae* amino acids using COOT. Finally, all fitted structures were rebuilt according to the cryo-EM map and guided by AlphaFold2 predictions of these subunits (*Jumper et al., 2021*; *Tunyasuvunakool et al., 2021*). APC9, APC13, APC15, the CDH1<sup>NTD</sup> and several loop regions not seen in previous structures were built *ab initio*, also guided by AlphaFold2 predictions of these subunits, including a prediction of the APC3:APC9 complex. Real-space refinement was performed in PHENIX (*Adams et al., 2010*) and the refined model was validated using the MolProbity tool (*Williams et al., 2018*). Maps and models were visualised with COOT (*Emsley and Cowtan, 2004*), and figures generated using Chimera X (*Goddard et al., 2018*). The refinement statistics are summarised in *Table 1*.

## Mass spectrometry

Purified proteins were prepared for mass spectrometric analysis by in solution enzymatic digestion, without prior reduction and alkylation. Protein samples were digested with trypsin or elastase (Promega), both at an enzyme to protein ratio of 1:20. The resulting peptides were analysed by nanoscale capillary LC-MS/MS using an Ultimate U3000 HPLC (Thermo Fisher Scientific Dionex) to deliver a flow of approximately 300nL/min. A C18 Acclaim PepMap100 5 µm, 100 µm Å~20 mM nanoViper column (Thermo Fisher Scientific Dionex), trapped the peptides before separation on a C18 Acclaim PepMap100 3 µm, 75 µm Å~250 mM nanoViper column (Thermo Fisher Scientific Dionex). Peptides were eluted with a 90 min gradient of acetonitrile (2–50%). The analytical column outlet was directly interfaced via a nano-flow electrospray ionization source, with a hybrid quadrupole orbitrap mass spectrometer (Q-Exactive Plus Orbitrap, Thermo Fisher Scientific). LC-MS/MS data were then searched against an in-house LMB database using the Mascot search engine (Matrix Science; *Perkins et al., 1999*), and the peptide identifications validated using the Scaffold program (Proteome Software Inc; *Keller et al., 2002*). All data were additionally interrogated manually.

## Acknowledgements

We are grateful to the LMB EM Facility for help with the EM data collection, J Grimmett, T Darling and I Clayson for computing, M Skehel and S Maslen for mass spectrometry and J Shi for help with insect cell expression. For the purpose of open access, the author has applied a CC BY public copyright license to any Author Accepted Manuscript version arising. This work was supported by UKRI/Medical Research Council MC_UP_1201/6 (D B) and Cancer Research UK C576/A14109 (D B).

## Additional information

### Funding

| Funder | Grant reference number | Author |
| --- | --- | --- |
| Cancer Research UK | C576/A14109 | Ester Vazquez-Fernandez Jing Yang Ziguo Zhang |
| Medical Research Council | MC_UP_1201/6 | David Barford |

The funders had no role in study design, data collection and interpretation, or the decision to submit the work for publication.

### Author contributions
Ester Vazquez-Fernandez, Conceptualization, Investigation, Methodology, Writing – review and editing; Jing Yang, Investigation; Ziguo Zhang, Antonina E Andreeva, Paul Emsley, Methodology; David Barford, Conceptualization, Resources, Supervision, Funding acquisition, Validation, Investigation, Writing - original draft, Writing – review and editing

### Author ORCIDs
David Barford (ID) https://orcid.org/0000-0001-8810-950X

Reviewer #1 (Public Review): https://doi.org/10.7554/eLife.100821.3.sa1
Reviewer #2 (Public Review): https://doi.org/10.7554/eLife.100821.3.sa2
Reviewer #3 (Public Review): https://doi.org/10.7554/eLife.100821.3.sa3
Author response https://doi.org/10.7554/eLife.100821.3.sa4

## Additional files

### Supplementary files
• Supplementary file 1. Tables of APC/C phosphorylation sites and ordered and disordered regions of APC/C subunits. (a) Table of CDK Phosphorylation sites of *S. cerevisiae* APC/C. (b) Ordered and disordered regions of *S. cerevisiae* APC/C subunits

• MDAR checklist

### Data availability
PDB and cryo-EM maps have been deposited with RCSB and EMDB respectively as accession numbers 8A5Y/EMD-15199, 8A61/EMD-15201 and 8A3T/EMD-15123 as listed in Table 1.

The following datasets were generated:

| Author(s) | Year | Dataset title | Dataset URL | Database and Identifier |
|---|---|---|---|---|
| Barford D, Fernandez-Vazquez E, Zhang Z, Yang J | 2022 | *S. cerevisiae* apo unphosphorylated APC/C | https://www.rcsb.org/structure/8A5Y | RCSB Protein Data Bank, 8A5Y |
| Barford D, Fernandez-Vazquez E, Zhang Z, Yang J | 2022 | *S. cerevisiae* apo phosphorylated APC/C | https://www.rcsb.org/structure/8A61 | RCSB Protein Data Bank, 8A61 |
| Barford D, Fernandez-Vazquez E, Zhang Z, Yang J | 2022 | *S. cerevisiae* APC/C-Cdh1 complex | https://www.rcsb.org/structure/8A3T | RCSB Protein Data Bank, 8A3T |
| Barford D, Fernandez-Vazquez E, Zhang Z, Yang J | 2022 | *S. cerevisiae* apo unphosphorylated APC/C | https://www.emdataresource.org/EMD-15199 | EMDataResource, EMD-15199 |
| Barford D, Fernandez-Vazquez E, Zhang Z, Yang J | 2022 | *S. cerevisiae* apo phosphorylated APC/C | https://www.emdataresource.org/EMD-15201 | EMDataResource, EMD-15201 |
| Barford D, Fernandez-Vazquez E, Zhang Z, Yang J | 2022 | *S. cerevisiae* APC/C-Cdh1 complex | https://www.emdataresource.org/EMD-15123 | EMDataResource, EMD-15123 |

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
