## [Editor Report · eLife assessment]

This study provides **compelling** data that defines the structure of the *S. cerevisiae* APC/C. The structure reveals overall conservation of its mechanism of action compared to the human APC/C but some **important** differences that indicate that activation by co-activator binding and phosphorylation are not identical to the human APC/C. Thus this study will be of considerable value to the field.

---

## [Referee Report · Reviewer #1 (Public Review)]

Summary:

This work focuses on the structure and regulation of the Anaphase-Promoting Complex/Cyclosome (APC/C), a large multi-subunit ubiquitin ligase that controls the onset of chromosome segregation in mitosis. Previous high-resolution structural studies have uncovered numerous structural features and regulatory mechanisms of the human APC/C, but it has remained unclear if these mechanisms are conserved in other model eukaryotes. To address this gap in our understanding, the authors employed cryo-electron microscopy to generate structural models of APC/C from the budding yeast *S. cerevisiae*, a key model organism in cell cycle analysis. In their comparison of the human and yeast complexes, the authors uncover many conserved structural features that are documented here in detail, revealing widespread similarities in the fundamental structural features of the enzyme. Interestingly, the authors also find evidence that two of the key mechanisms of human APC/C regulation are not conserved in the yeast enzyme. Specifically:

(1) The ubiquitin ligase activity of the APC/C depends on its association with a co-activator subunit such as CDH1 or CDC20, which serves both as a substrate-binding adaptor and as an activator of interactions with the E2 co-enzyme. Previous studies of the human APC/C revealed that co-activator binding induces a conformational change that enables E2 binding. In contrast, the current work shows that this E2-binding conformation already exists in the absence of a co-activator in the yeast enzyme, suggesting that the enhancement of E2 binding in yeast depends on other, as yet undiscovered, mechanisms.

(2) APC/C phosphorylation on multiple subunits is known to enhance APC/C activation by the CDC20 co-activator in mitosis. Previous studies showed that phosphorylation acts by promoting the displacement of an autoinhibitory loop that occupies part of the CDC20-binding site. In the yeast enzyme, however, there is no autoinhibitory loop in the CDC20-binding site, and there is no apparent effect of APC/C phosphorylation on co-activator binding sites. Thus, phosphorylation activates the yeast CDC20-APC/C by unknown mechanisms.

Strengths:

The strength of this paper is that it provides a comprehensive analysis of yeast APC/C structure and how it compares to previously determined human structures. The article systematically unwraps the key features of the structure in a subunit-by-subunit fashion, carefully revealing the key features that are the same or different in the two species. These descriptions are based on a thorough overview of past work in the field; indeed, this article serves as a concise review of the key features, conserved or otherwise, of APC/C structure and regulation.

Weaknesses:

No significant weaknesses were identified.

---

## [Referee Report · Reviewer #2 (Public Review)]

Summary:

This paper from the Barford lab describes medium/high-resolution cryo-EM structures of three versions of the *S. cerevisiae* anaphase-promoting complex/cyclosome (APC/C):

(1) the recombinant apo complex purified from insect cells,

(2) the apo complex phosphorylated in vitro by cyclin-dependent kinase, and

(3) an active APC/C-Cdh1-substrate ternary complex.

The focus of the paper is on comparing similarities and differences between *S. cerevisiae* and human APC/C structures, mechanisms of activation by coactivator, and regulation by phosphorylation. The authors find that the overall structures of *S. cerevisiae* and human APC/C are remarkably similar, including the binding sites and orientation for the substrate-recruiting coactivator, Cdh1. In addition, the mechanism of Cdh1 inhibition by phosphorylation appears conserved across kingdoms. However, key differences were also observed that reveal divergence in APC/C mechanisms that are important for researchers in this field to know. Specifically, the mechanism of APC/C-Cdc20 activation by mitotic phosphorylation appears to be different, due to the absence of the key Apc1 autoinhibition loop in the *S. cerevisiae* complex. In addition, the conformational activation of human APC/C by coactivator binding was not observed in the *S. cerevisiae* complex, implying that stimulation of E2 binding must occur via a different mechanism in this species.

Strengths:

Consistent with the numerous prior cryo-EM structures of human APC/C from the Barford lab, the technical quality of the structure models is a major strength of this work. In addition, the detailed comparison of similarities and differences between the two species will be a very valuable resource for the scientific community. The manuscript is written very well and allows readers lacking expertise in cryo-EM to understand the important aspects of the conservation of APC/C structure and mechanism across kingdoms.

Weaknesses:

The lack of experimentation in this work to test some of the putative differences in APC/C mechanism (e.g. stimulation of E2 binding by coactivator and stimulation of activity by mitotic phosphorylation) could be considered a weakness. Nonetheless, the authors do a nice job explaining how the structure interpretations imply these differences likely exist, and this work sets the stage nicely for future studies to understand these differences at a mechanistic level. There is enough value in having the *S. cerevisiae* structure models and the comparison to the human structures, without any additional experimentation.

The validation of APC/C phosphorylation in the unphosphorylated and hyperphosphorylated states is not very robust. Given the lack of significant effects of phosphorylation on APC/C structure observed here (compared to the human complex), this becomes important. A list of phosphorylation sites identified by mass spec before and after in vitro phosphorylation is provided but lacks quantitative information. This list indicates that a significant number of phosphorylation sites are detected in the purified APC/C prior to reaction with purified kinases. Many more sites are detected after in vitro kinase reaction, but it isn't clear how extensively any of the sites are modified. There is reason for caution then, in accepting the conclusions that structures of unphosphorylated and hyperphosphorylated APC/C from *S. cerevisiae* are nearly identical.

---

## [Referee Report · Reviewer #3 (Public Review)]

Vazquez-Fernandez et al. present a comprehensive and detailed analysis of the *S. cerevisiae* APC/C complex, providing new insights into its structure and function. The authors determined the medium-resolution structures of three recombinant *S. cerevisiae* APC/C complexes, including unphosphorylated apo-APC/C (4.9 Å), the ternary APC/CCDH1-substrate complex (APC/CCDH1:Hsl1 , 4.0 Å), and phosphorylated apo-APC/C (4.4 Å). Prior structures of human, E. cuniculi, *S. cerevisiae*, and *S. pombe* APC/C subunits, as well as AlphaFold2 predictions were used to guide model building. Although the determined structures are not sufficient to fully explain the molecular mechanism of APC/C activation and regulation in S. cerevisiae, they provide valuable insights into the similarities and differences with the human complex, shedding light on the conserved and divergent features of APC/C function.

The manuscript synthesizes the structural analysis of the APC/C complex in *S. cerevisiae*, with literature into a cohesive and clear picture of the complex's structure and function. It is well-written and clear, making the complex biology of the APC/C complex accessible to a wide range of readers. The complex forms a triangular shape, with a central cavity surrounded by two modules: the TPR lobe and the platform module. The TPR lobe consists of three TPR proteins (APC3, APC6, and APC8), which stack on top of each other to form a quasi-symmetric structure. The platform module is composed of the large APC1 subunit, together with APC4 and APC5. The authors also analyzed the structure of several smaller subunits that are involved in regulating the activity of the APC/C complex and showed their structural similarities to and discrepancies from their human counterparts. These subunits, including CDC26/APC12, SWM1/APC13, APC9, and MND2/APC15, form extended, irregular structures that simultaneously contact multiple large globular APC/C subunits.

While the authors report the similarity between the overall structure of *S. cerevisiae* and human APC/C complexes, they also found two unexpected differences. First, in the *S. cerevisiae* apo-complex, the E2 binding site on APC11RING is accessible, whereas, in humans, it requires CDH1 binding. Second, a structural element similar to the human APC1 auto-inhibitory segment is missing in *S. cerevisiae*. In humans, the phosphorylation-dependent displacement of this segment allows CDC20 binding to APC/C. In *S. cerevisiae*, the binding requires phosphorylation however the structures reported here are suggestive that this could involve a different (presently unknown) mechanism. These structural insights highlight the importance of understanding the species-specific features of APC/C function.

Strengths:

The manuscript does a great job of revealing new structures.

Opportunity for increasing impact: It would have been nice if some functional differences were demonstrated, for example regarding the mechanism of CDC20 binding, and the comparison between apo-APC/C and ternary APC/CCDH1:Hsl1 does not explain the molecular activation mechanism of *S. cerevisiae* APC/C. Nonetheless, the authors nicely integrate their data with well-established literature on the similarities and differences between yeast and human systems.

---

## [Author Response]

The following is the authors’ response to the original reviews.

**Reviewer #1 (Recommendations For The Authors):**
(1) The introduction includes the following sentence: "CDH1 interacts with the APC/C during G1 and S-phase. On entering mitosis, CDK and polo kinases phosphorylate the APC/C and CDH1 to effect switching to CDC20." In fact, CDH1 is inactivated from late G1 to mid-mitosis as a result of phosphorylation by G1/S, S phase, and mitotic Cdk-cyclin complexes. Phosphorylation of the APC/C, not inactivation of CDH1, enables the switch to CDC20.

Thank you for this. We have corrected the text and include the Zachariae et al (1998) reference.

(2) Supplementary Table 1 provides a long list of APC/C sites phosphorylated in vitro by Cdk2-cyclin A-Cks2 and Plk1 (note that the main text states only Cdk2-cyclin A). It seems likely that the high amount of kinase in these reactions has led to minor levels of phosphorylation at many of these sites. Although I realize that these results are peripheral to the central findings of the paper, it would be helpful to see confidence scores or other evidence of significance for the indicated phosphopeptides. Perhaps the Cdk consensus sites could be marked on the table in some way, and a description of the MS methods could be provided in the Methods section.

We have implemented this useful suggestion to highlight the Cdk consensus sites. Unfortunately we don’t have confidence scores of significance of the indicated phosphopeptides.

**Reviewer #2 (Recommendations For The Authors):**
(1) My only real concern is with the phosphorylated APC/C structure. The authors provide a table that lists a bunch of phosphorylation sites detected before and after in vitro phosphorylation by purified kinases, and in the purified protein gels, some mobility shifts that would be consistent with significant phosphorylation are observed for some of the subunits. However, the mass spec data are non-quantitative. It would be more useful to provide estimated stoichiometries for the various phosphorylation sites to help support the expectation that the complex is heavily phosphorylated and that the structure presented actually represents hyperphosphorylated APC/C. No evidence of phosphorylated amino acids is noted in the cryo-EM structures, presumably because resolution is not high enough and/or there is too much flexibility in these areas. Given that hyperphosphorylation does not affect enzymatic activity and has very little impact on the complex structure, it seems important to provide readers with additional confidence that the complex is indeed heavily phosphorylated and that the complex isolated from insect cells is not heavily phosphorylated. Since the complex is purified from a eukaryotic expression system it seems formally possible that key phosphorylation sites could already be present due to the activity of endogenous Cdk or other kinases in insect cells and, indeed, quite a few sites are noted to exist even without in vitro phosphorylation. Providing stoichiometry of these sites might help address the likelihood that key regulatory sites are already occupied upon purification. It might at least be worth addressing this in the text.

The suggestion to comment on the level of phosphorylation of the ‘unphosphorylated’ and Cdk-treated ‘phosphorylated’ APC/C is an excellent idea. The text has been modified on page 9 to include such a discussion. Unfortunately we don’t have quantification of the stoichiometry of the phospho-sites.

(2) On a minor note, in the results text the authors mention that cyclin A-Cdk2 is used for in vitro phosphorylation, but in the methods, it states both cyclin A-Cdk2 and Plk1 are used. This should be edited for consistency.

Thank you for noticing this. Now corrected.

(3) Another minor issue - the authors state in the introduction (third paragraph) that "CDH1 interacts with the APC/C during G1 and S-phase". Actually, Cdh1 becomes phosphorylated and APC/CCdh1 inactivated at S-phase onset, both in *S. cerevisiae* and humans. In fact, phosphorylation of Cdh1 is an important driver of the irreversible transition from G1 to S-phase. This statement should be corrected.

Thank you for noticing this. This error has been corrected and include the Zachariae et al (1998) reference.

**Reviewer #3 (Recommendations For The Authors):**
To be addressed in a revised manuscript:(1) The authors should cite and discuss Cole Ferguson et al., Mol Cell 2022. This study describes the loss of APC7 in a human disease and provides a detailed structural and biochemical examination of the effects of APC7 loss on human APC/C. Given that much of our understanding of APC7 comes from this work, it should be highlighted in the introduction and discussed in depth in light of the new work on *S. cerevisiae* APC/C.

Thank you for mentioning this interesting paper. We discuss its main findings in the ‘Discussion’. Given the paper shows that deletion of APC7 has no discernible effect on the stability of human APC/C, we have deleted the discussion that APC7 stabilises human APC/C analogous to the stabilisation conferred on *S. cerevisiae* APC/C by APC9.

(2) There are multiple cases in the manuscript where the text was referring to the human complex but APC/CCdh1:Hsl1 was written, including labeling of Figure 4b. It would be useful to consider nomenclature considering that Hsl1 is a yeast protein.

Thank you for noticing this. We mistakenly wrote ‘Hsl1’ instead of ‘Emi1’. Now corrected.

(3) The authors should tone down claims regarding their discoveries absence of APC7 in *S. cerevisiae*. The absence of APC7 has been known for nearly two decades and the authors confirm this (Pan et al.l 2007, Journal of Cell Science) and then show the structure.

We agree with this as explained in response to point 1.

(4) On page 7, the authors are writing about the four helices mediating the APC/C-CDH1 interactions but list only 3.

We have revised the sentence to clarify this point.